# Toxic Effects of Glyphosate on the Nervous System: A Systematic Review

**DOI:** 10.3390/ijms23094605

**Published:** 2022-04-21

**Authors:** Carmen Costas-Ferreira, Rafael Durán, Lilian R. F. Faro

**Affiliations:** Department of Functional Biology and Health Sciences, Faculty of Biology, Universidade de Vigo, Campus Lagoas-Marcosende, 36310 Vigo, Spain; maica.cf@hotmail.com (C.C.-F.); rduran@uvigo.es (R.D.)

**Keywords:** glyphosate, glyphosate-based herbicides (GBH), neurotoxic effects, human, rodent, fish

## Abstract

Glyphosate, a non-selective systemic biocide with broad-spectrum activity, is the most widely used herbicide in the world. It can persist in the environment for days or months, and its intensive and large-scale use can constitute a major environmental and health problem. In this systematic review, we investigate the current state of our knowledge related to the effects of this pesticide on the nervous system of various animal species and humans. The information provided indicates that exposure to glyphosate or its commercial formulations induces several neurotoxic effects. It has been shown that exposure to this pesticide during the early stages of life can seriously affect normal cell development by deregulating some of the signaling pathways involved in this process, leading to alterations in differentiation, neuronal growth, and myelination. Glyphosate also seems to exert a significant toxic effect on neurotransmission and to induce oxidative stress, neuroinflammation and mitochondrial dysfunction, processes that lead to neuronal death due to autophagy, necrosis, or apoptosis, as well as the appearance of behavioral and motor disorders. The doses of glyphosate that produce these neurotoxic effects vary widely but are lower than the limits set by regulatory agencies. Although there are important discrepancies between the analyzed findings, it is unequivocal that exposure to glyphosate produces important alterations in the structure and function of the nervous system of humans, rodents, fish, and invertebrates.

## 1. Introduction

Glyphosate, N-(phosphonomethyl) glycine is the most widely used herbicide in the world [1]. It is a non-selective systemic biocide with broad-spectrum activity and was introduced in 1974 for the control of weeds in agricultural production fields [2]. The widespread use of glyphosate in agriculture and forestry has contributed to the development of numerous commercial formulations containing this compound. Herbicide formulations containing this active ingredient represent approximately 60% of the global market for non-selective herbicides [3].

In recent years, the use of glyphosate has spread worldwide due to the development of glyphosate-resistant crops. The main driver of the enormous success of this technology has been the economic benefits obtained in the agricultural sector after the introduction of genetically modified crops [4]. This has contributed to the positioning of glyphosate-based herbicides (GBH) as leaders in the global pesticide market. Thus, around 600.000 to 750.000 tons of glyphosate are used each year, and it is estimated that its use will increase reaching between 740.000 and 920.000 tons by 2025 [5].

The mechanism of action of glyphosate is associated with its ability to block the shikimic acid pathway, which is involved in the synthesis of aromatic amino acids in plants, fungi, and some microorganisms [6]. Glyphosate inhibits the enzyme 5-enolpyruvilshikimate-3-phosphate synthase, which is the penultimate step in the shikimate pathway [7]. This inhibition leads to a reduction in the synthesis of the aromatic amino acids tyrosine, phenylalanine, and tryptophan, as well as a decrease in protein synthesis [8] (Figure 1). Therefore, blocking this metabolic pathway eventually causes the death of the target organism within a few days [9].

The absence of the shikimate pathway in animals has led to the conclusion that GBH does not pose a health risk to animals and humans [10]. Moreover, many investigations on glyphosate toxicity in animals have suggested the low toxicity of this compound, the adverse effects of which have only been observed after exposure to relatively high doses [8,11,12]. These data led to the classification of glyphosate in the least toxic category (category IV, practically non-toxic and non-irritating) by the United States Environmental Protection Agency (EPA) [13,14].

In general, in the different environmental compartments, glyphosate is mainly degraded by microorganisms, so that its persistence is considered to be low to moderate, although it is considerably variable. On the one hand, although glyphosate is assumed to be readily degraded in soil, its biodegradation is influenced by numerous factors, including physico-chemical, biological properties and soil composition [15]. Thus, the half-life of glyphosate in soil can range from 1 to 280 days, while that of aminomethylphosphonic acid (AMPA), its main metabolite, ranges from 23 to 958 days [16,17,18]. In soil, glyphosate can bind strongly to its constituent particles and remain biologically inactive, or it can reach groundwater, due to its high water solubility [19]. However, repeated applications of glyphosate have been shown to result in a gradual difficulty for its biodegradation in the soil, which could increase the risk of groundwater contamination [20,21]. In water, the permanence of glyphosate is also widely variable and depends on factors such as light and temperature, being more persistent and toxic under conditions of darkness and higher water temperatures [22]. In general, the half-life of glyphosate in water varies from a few days to 91 days [23,24], although it has been found to remain for up to 315 days in marine waters [25]. On the other hand, while the persistence of glyphosate in vegetation may be only days, several studies have detected its presence in many foods and crops even a year after application [26,27].

Therefore, although the concentrations of glyphosate residues that persist over time are relatively low, it is possible that due to extensive use on a large scale they may accumulate and become a risk to animal and human health, as they are chronically exposed to residues in the water and food they consume [23,28,29]. This has been confirmed by the detection of glyphosate in the organs and urine of a high proportion of farm animals and farmers [30,31,32,33]. In addition, residues were also found in the urine of 60–80% of the general population in the United States at medium and maximum concentrations of 2–3 and 233 μg/L, respectively. In Europe, residues were also detected in the urine of 44% of the population, although their average and maximum concentrations were lower: <1 and 5 μg/L, respectively [9,28].

Recently, data on glyphosate contamination in the environment suggest that acute toxicity may not be as relevant as toxicity from chronic exposure to lower concentrations of this compound. Therefore, the number of publications demonstrating the chronic toxicity of glyphosate in animals and humans has increased considerably. This has led to increased concern about the potential harmful side effects that chronic exposure to glyphosate could have on animal and human health. Glyphosate has recently received increasing attention from the scientific community and from national and international regulatory agencies. Based on research on the chronic side effects of glyphosate, in 2015, the International Agency for Research on Cancer (IARC) of the World Health Organization (WHO) reclassified glyphosate as probably carcinogenic to humans. Nevertheless, IARC’s conclusion has not been confirmed by European Union assessment or the recent joint assessment by the Food and Agriculture Organization of the United Nations (FAO)/OMS [34]. This highlights the existence of significant disagreement between regulatory agencies, as well as the need to reach consensus and update safety standards for glyphosate in order to protect public health and the environment.

Numerous commercial formulations of glyphosate contain several adjuvants, which improve the penetration of the active ingredient into the target plants and increase their efficacy [35,36]. It has been postulated that the activity of GBH is not exclusively due to the active ingredient but could be due to the intrinsic toxicity of the adjuvants or the possible synergy between glyphosate and the other ingredients of the formulation [37]. In fact, polyethoxylated tallow amine, the predominant surfactant in several commercial formulations, has been found to increase glyphosate-induced toxicity by facilitating its penetration through plasma membranes [38,39,40,41,42]. Therefore, it is important for research to evaluate the toxicity of both commercial formulations and pure glyphosate [43].

Taken together, above information shows that the intensive and widescale use of glyphosate can constitute a major environmental and health problem. In recent years, there has been an increase in the number of scientific publications dealing with the possible side effects of glyphosate and its commercial formulations. Nevertheless, a full review of research articles on the effects of glyphosate on the nervous system is still needed.

Therefore, in the present study, we carry out a systematic review of the scientific literature available on the effects and mechanisms of action of glyphosate and its commercial formulations on the nervous system of various animal species and humans. The main objective of this review is to understand the risks arising from increasing exposure to glyphosate residues in the environment and in food.

## 2. Methodology

The present review was carried out with the aim of unifying the results of the most recent studies on the effects of glyphosate and its commercial formulations on animal health. For this purpose, a systemic review was performed following the guidelines established by the preferred reporting items for systematic reviews and meta-analyses (PRISMA) [44]. Searches were carried out in the specialized databases PubMed, Scopus, and Web of Science in February 2022, with a time restriction limited to studies published in the last ten years.

The following terms referring to glyphosate and its commercial formulations were used to select the scientific articles to be included in this review: “glyphosate”, “Roundup”, and “GBH”. Each of the three terms mentioned above was entered into the databases according to the following search strategies: “((glyphosate OR Roundup OR GBH) AND (nervous system))”; “((glyphosate OR Roundup OR GBH) AND (neurotoxicity))”; “((glyphosate OR Roundup OR GBH) AND (nervous system) AND (effects))”; ((glyphosate OR Roundup OR GBH) AND (nervous system) AND (toxicity))”.

### Exclusion and Inclusion Criteria

The articles included in this review met the following inclusion criteria: (1) original studies in article format; (2) in English or Spanish; and (3) studying the effects of the pesticide on the nervous system. Articles were excluded according to the following exclusion criteria: (1) theoretical articles or reviews; (2) studies in which glyphosate was administered in combination with other pesticides; (3) studies evaluating the effects of pesticides other than glyphosate; (4) case studies; and (5) studies that used glyphosate doses above the no-observed-adverse-effect levels (NOAEL) established by regulatory agencies.

As a result of the searches carried out in the three databases, 922 articles published in the last ten years were identified. The articles were then exported to Refworks to eliminate duplicates. In accordance with the inclusion and exclusion criteria, 163 titles and abstracts were screened to verify whether they met the previously mentioned criteria. After this procedure, 112 articles were excluded for the reasons summarized in Figure 2. Ultimately, 51 articles were included in the present systematic review.

## 3. Results

### 3.1. Effects of Glyphosate in Humans

A series of studies show that glyphosate and its commercial formulations can produce detrimental effects on the human nervous system. These investigations have shown that glyphosate can cross and affect the blood–brain barrier (BBB) and cause various types of short-term or long-term disturbances in the human nervous system (Table 1).

#### 3.1.1. Descriptive and Analytical Studies

Although most of studies in humans mainly describe the consequences of glyphosate poisoning after suicide attempts, it appears that occupational or chronic exposure to this pesticide (via inhalation and dermal routes) may also cause neurotoxic effects. In a study by Fuhrimann et al. [45], the authors describe that glyphosate exposure has been associated with the development of visual memory impairment in Ugandan smallholder farmers. However, other studies have not found an association between occupational exposure to glyphosate and increased risk of health problems, such as nerve conduction abnormalities [48,49]. These results have led some authors to postulate that glyphosate is less toxic to farmers’ health than other pesticides. Therefore, future research is needed to follow up and compare the potential toxic effects that agricultural use of different pesticides may exert on human health.

Another consideration about the effects of glyphosate is the fact that its effects do not appear immediately after exposure but one or two days later. In this sense, Lee et al. [46] found that S100 calcium-binding protein B (S100B) could be an important predictor of neurological complications in patients poisoned with glyphosate because its levels were increased in the group that was exposed to this substance and that presented neurological alterations. S100B levels peaked on the second day after exposure, indicating that glyphosate can reach the brain parenchyma and cause maximum brain damage after some time [46].

#### 3.1.2. In Vitro Studies with Human Line Cells

The ability of glyphosate to cross the BBB was also reported in an in vitro study by Martínez and Al-Ahmad [51]. In this study, the authors observed that both glyphosate and its metabolite AMPA can increase BBB permeability, possibly by interfering with the proteins that mediate the hermetic junctions between the endothelial cells that comprise the BBB. This study also showed that glucose uptake by brain endothelial cells increased after exposure to high doses of glyphosate [51]. Glucose is the main source of energy for the brain, and its entry into the central nervous system (CNS) is mediated by microvascular endothelial cells. Therefore, increased availability of glucose after exposure to glyphosate could alter the metabolic activity of neurons, as seen in a study by Martínez and Al-Ahmad [51]. This investigation also documented a decrease in cellular metabolism that did not appear to be related to neurotoxicity, as neurite density and formation were not affected by treatment with glyphosate.

Exposure to glyphosate appears to affect neuronal development in the human CNS, altering the expression of molecules involved in the growth and maturation of neurons. GBH administration has been shown to alter the proliferation of cells in culture, although this effect did not occur when glyphosate was administered alone [50]. In addition, exposure to glyphosate and its metabolite induced a negative regulation in the expression of the *TUBB3* and *GAP43* genes, which are responsible for the synthesis of neuronal cytoskeleton proteins and axonal growth cones, respectively [52].

Likewise, exposure to glyphosate induced an increase in the levels of mRNA-Wnt3a, -Wnt5a, and -Wnt7a, which are also related to the regulation of neuronal development [52]. Given that the components of the Wnt signaling pathways are expressed in a strictly controlled manner during development, their deregulation by glyphosate could promote neuronal morphological defects and cause changes in the correct neuronal development [53,54].

In line with this, there is evidence of the participation of Wnt signaling in various neurocognitive developmental disorders, such as autism [55,56]. Thus, the deregulation of this pathway by glyphosate in human cells in vitro could be related to a higher incidence of developmental and autism spectrum disorders in children whose mothers were exposed to pesticides [57,58,59], including glyphosate, during pregnancy [47].

Martínez and colleagues [52] also showed that glyphosate increased the mRNA expression of the two isoforms of calcium-calmodulin-dependent protein kinase 2 (CAMK2A and CAMK2B). The product of these genes appears to have a dual role, as, although they are related to neuronal development and survival, they also contribute to regulation of neuronal death in response to a variety of insults [60,61]. Thus, the increased expression of this glyphosate-induced mRNA could be related to neuronal apoptosis triggered in response to oxidative stress. However, according to a study by Martínez et al. [52], AMPA decreased the expression levels of this mRNA, which led the authors to suggest that glyphosate metabolism could partially prevent neuronal death.

Another effect observed in a study by Martínez et al. [52] is an increase in oxidative stress, evidenced as an increase in the production of reactive oxygen species (ROS) and nitric oxide (NO), as well as lipid peroxidation (LPO). In addition, glyphosate and its metabolite also potentiated an inflammatory response by upregulating the expression of the proinflammatory cytokine interleukin 6 (IL-6) genes and tumor necrosis factor-alpha (TNF-α).

As mentioned previously, injuries caused by glyphosate, such as neuroinflammation or oxidative stress, can cause neuronal death. It was shown that both glyphosate and AMPA reduced the viability of human cells and increased the leakage of lactate dehydrogenase (LDH) [52]. LDH is involved in energy production, is found in almost every organ in the body (including the brain), and, when an organ or tissue is damaged, is released into the blood. Thus, glyphosate-induced LDH increases could be indicative of damage caused by the herbicide in the CNS. Likewise, there was also an upregulation in the gene expression of the different pathways of cell death (apoptosis, autophagy, and necrosis), including caspases 3 and 7, involved in the execution of apoptosis [52]. These results corroborate the possible involvement of mechanisms of apoptosis, autophagy, and necrosis in glyphosate-induced cell death in humans.

### 3.2. Effects of Glyphosate in Rodents

Several studies show that exposure to glyphosate or GBH produces many toxic effects on both the CNS and peripheral nervous system (PNS) of rodents. The main effects observed include changes in the development of the nervous system and in the neurotransmission systems, as well as oxidative stress and neuroinflammation, processes that lead to neuronal death and the appearance of behavioral changes (Table 2 and Table 3). Most of these studies show the neurotoxic effects of glyphosate administered at early ages during the intrauterine period and lactation, although chronic or acute exposure in adulthood also causes important alterations in the function and structure of the nervous system.

#### 3.2.1. Development of Nervous System

Data from in vitro and ex vivo studies show that exposure to glyphosate in the early stages of neuronal development induces dysregulation of various signaling pathways and cascades, leading to delayed neuronal differentiation, growth, migration, and myelination processes in both the CNS and PNS during this period [76,78,79]. Coullery et al. [67] showed that glyphosate downregulates the Wnt5a/CaMKII signaling pathway in embryonic hippocampal neurons, a cascade that controls neuronal circuit formation and integration. Similarly, Luna et al. [74] found that glyphosate administered in the early postnatal stage reduced the expression of CAMKII in the hippocampus. Because of this deregulation, glyphosate would alter the process of neuronal differentiation and the subsequent formation of synaptic connections between hippocampal neurons [80]. This was verified by the team of Luna et al. [74], who observed that glyphosate administration was associated with a decrease in dendritic complexity and synapse formation in hippocampal neurons. However, Cattani et al. [37] observed a different effect, as GBH exposure activated the CaMKII pathway. These authors related CaMKII activation to the increase in glutamate-induced Ca^2+^ influx in the hippocampus of GBH-treated animals. The difference between studies may be related to the time of exposure, as Coullery et al. [67] and Luna et al. [74] applied a subchronic treatment with glyphosate, whereas Cattani and colleagues [37] applied a single exposure to GBH in vitro.

It was also observed that GBH downregulated the expression of brain-derived neurotrophic factor (BDNF) in the prefrontal cortex (PFC) and hippocampus [64]. BDNF plays an essential role in the processes of neurogenesis, growth, survival, and neuronal plasticity. In the brain, the BDNF binds to tyrosine-related kinase receptor B (TrkB) and activates an intracellular signaling cascade to promote synaptic survival and plasticity [81,82]. In this regard, GBH exposure upregulated TrkB expression in the PFC, which could reflect a compensatory mechanism to balance the reduction in BDNF levels [64].

Exposure to GBH also modified the expression of S100B. In vitro exposure to GBH induced a downregulation of this protein during development, but its levels increased in adult offspring after chronic exposure [3]. The S100B protein is mainly found in the glial cells (predominantly astrocytes) and exerts trophic or toxic effects, depending on its concentration [83]. At nanomolar concentrations, S100B acts as a neurotrophic factor that promotes neuronal growth and survival in the developing or injured nervous system. Conversely, at micromolar levels, S100B is thought to stimulate the expression of inflammatory cytokines that cause cell death. Thus, increased levels of this protein are considered a marker of nervous system damage. Down-regulation of S100B levels in the early stages of development could reflect one of the mechanisms by which glyphosate alters proper neuronal development, whereas upregulation of this protein during adulthood could be indicative of damage to the CNS induced by glyphosate.

Nuclear factor kappa B (NF-kB) expression is another parameter modified after GBH exposure [3]. NF-kB proteins represent a family of transcription factors that are expressed in both neuronal and non-neuronal cells. These proteins are involved in a wide variety of functions, including differentiation, cell survival, synaptic plasticity, or adult neurogenesis, among many others. Exposure to glyphosate during the periods of pregnancy and lactation induced a decrease in NF-kB activation in the hippocampus of immature rats, indicating that early exposure to this compound could affect neurogenesis and therefore the correct development of CNS [3].

Likewise, Ji et al. [71] demonstrated that GBH exposure modified the expression of numerous microRNAs that are implicated in brain development and in the pathogenesis of various diseases. MicroRNAs are small, non-coding RNAs that are involved in gene silencing and thus can shape the landscape of post-transcriptional gene expression [84]. They are also involved in numerous biological processes, such as early neurogenesis, circuit development, or synaptic plasticity, and their dysregulation is linked to the onset of numerous diseases. In an in vitro study, Masood et al. [76] observed the ability of glyphosate to modify gene expression, as its administration negatively regulated the expression of the cytoprotective *CYP1A1* gene. These results reflect the fact that altered gene expression patterns could be another mechanism underlying glyphosate-induced neurotoxicity.

GBH-induced neurotoxicity in the hippocampus has also been linked to the recruitment of signal transduction pathways, leading to the activation of kinase cascades, including extracellular signal-regulated kinase (ERK) [3,37]. The ERK cascade is involved in the regulation of a wide variety of cellular functions, such as proliferation, differentiation, neuronal survival, and synaptic plasticity [85,86]. However, previous studies have shown that ERK activation can also mediate cell death [87,88]. Specifically, Jiang et al. [89] and Satoh et al. [90] found that activation of this pathway was involved in glutamate-induced neuronal apoptosis. Given that in a study by Cattani et al. [37], it was shown that GBH, in addition to ERK overactivation, also increased glutamate levels, the authors linked the activation of this pathway to increases in glutamate levels and subsequent neuronal death by apoptosis.

The glutamate excitotoxicity and calcium overload observed by Cattani et al. [37] may be related to the decrease in the expression of proteins of the dynorphin family observed later by the same authors [66]. This is because dynorphins are opioid receptor agonists [91], but they can also have other effects, such as the modulation of N-methyl-D-aspartate glutamatergic receptors (NMDAR) in the hippocampus and voltage-gated calcium channels (VDCCs) [92,93,94], and a decrease in their levels can trigger the previously observed effects on neurotransmission. Furthermore, dynorphins act as protectors of dopaminergic neurons and reduce the progression of Parkinson’s disease [95]. Thus, the decrease in dynorphin levels observed by Cattani et al. [66] in the substantia nigra could also be related to increased vulnerability of dopaminergic neurons to environmental pollutants, such as glyphosate.

The previously mentioned alterations could be implicated in the delay in sensorimotor development observed after GBH exposure in a study by Ait-Bali et al. [64]. Specifically, GBH exposure caused a delay in the development of innate reflexes and a deficit in the motor development of the offspring.

However, glyphosate not only appears to affect neural development in immature offspring but could also alter neurogenesis during adulthood. Various brain regions in adult females undergo remodeling during the peripartum period to adapt their behavior to the needs of the offspring [96,97]. One such region is the hippocampus, which has received much attention due to its persistent ability to generate new neurons even in adulthood [98]. It has been shown that glyphosate alone or in formulation affected aspects of neurogenesis in the maternal hippocampus during the postnatal period [68].

#### 3.2.2. Effects on Neurotransmission

Neurotransmission is another fundamental process in the functioning of the nervous system on which glyphosate seems to exert a toxic effect, the glutamatergic system being one of the most affected after exposure to the pesticide. Glutamate is the most abundant excitatory neurotransmitter in the nervous system and is involved in various cognitive functions, such as learning and memory. Under physiological conditions, glutamate is maintained mainly in the intracellular medium, and only a small fraction exists outside the cells, as an increase in the extracellular levels of this neurotransmitter and its interaction with pre- or postsynaptic receptors can induce neuronal death by excitotoxicity [99,100,101].

Studies by Ait-Bali et al. [64] and Cattani et al. [3,37] show the effects of early GBH exposure on glutamatergic neurotransmission in immature or adult offspring. Acute or chronic treatment with GBH increased in vitro glutamate release, decreased its reuptake by astrocytes, and increased Ca^2+^ influx into hippocampal terminals. Furthermore, GBH increased the expression of NMDAR in both the hippocampus and the PFC [64] and activated these receptors and L-type voltage-dependent calcium channels (L-VDCC) in the hippocampus [3,37]. On the other hand, it was also demonstrated that GBH may alter glutamate metabolism by decreasing the activity of glutamic-pyruvic transaminase (GPT) and glutamic-oxaloacetic transaminase (GOT) [69]. Furthermore, an increase in glutamine transport into neurons was observed after chronic GBH exposure [37].

On the other hand, both early and late exposure to glyphosate seems to also affect cholinergic neurotransmission. Acetylcholine is a neurotransmitter that plays a central role in learning, attention, and synaptic plasticity [102,103]. In a study by Gallegos et al. [69], it was shown that GBH exposure reduced the number of cholinergic neurons in the medial septum, which was evidenced by a decrease in the expression of choline acetyltransferase (ChAT), the enzyme responsible for the synthesis of acetylcholine. GBH also reduced levels of the α7-type nicotinic acetylcholine receptor in the hippocampus. These data suggest that exposure to glyphosate alters the functioning of the septo-hippocampal cholinergic pathway, which is involved in the processing of memory [104,105,106].

Several of the studies reviewed here show that glyphosate or GBH was able to induce a slight decrease in acetylcholinesterase (AChE) activity in different brain areas [3,63,64,69,73]. The effect of glyphosate on AChE activity differs considerably depending on treatment and brain area analyzed, but in all cases, it is observed that, unlike other organophosphate pesticides, glyphosate is a weak inhibitor of AChE. Therefore, concentrations much higher than ambient levels of glyphosate that would be required to produce an effective inhibition of the activity of this enzyme in the brain [73].

Another neurotransmitter system affected by glyphosate exposure is the dopaminergic system. Dopamine is a biogenic amine involved in the modulation of locomotor activity, affectivity, and neuroendocrine communication in the CNS, and its alteration can lead to the development of neurodegenerative disorders, such as Parkinson’s disease [106]. Studies by Ait-Bali et al. [62,64], Hernández-Plata et al. [70], and Martínez et al. [13] show the effects of glyphosate on mesocorticolimbic and nigrostriatal dopaminergic neurotransmission. Ait-Bali et al. [62,64] have shown that early or adult exposure to GBH caused a decrease in the number of dopaminergic neurons, observed as a decrease in immunoreactivity for the enzyme tyrosine hydroxylase (TH) in the substantia nigra pars compacta and ventral tegmental area.

Furthermore, in studies by Hernández-Plata et al. [70] and Martínez et al. [13], the authors documented that systemic administration of glyphosate in adult rats significantly decreased total dopamine content, especially in the striatum, PFC, and hippocampus, in addition to increasing dopamine turnover in the PFC and hippocampus. Likewise, Hernández-Plata et al. [70] demonstrated that glyphosate decreased extracellular levels of dopamine and its metabolites in the striatum, as well as the specific binding of dopamine to its D1-type receptor in the nucleus accumbens. Thus, glyphosate could affect mechanisms that regulate the density or affinity of dopaminergic D1 receptors. However, these alterations in dopaminergic neurotransmission appear to occur when a critical concentration of glyphosate is present in the system but disappear over time. Taken together, the results of these studies seem to indicate that glyphosate exposure produces short-term effects on dopaminergic neurotransmission in various brain regions.

Other alterations in neurotransmitter processes documented after exposure to glyphosate or GBH include a decrease in the number of serotonergic neurons and in the contents of noradrenaline and serotonin in various brain areas [13,62]. Both serotonin and noradrenaline play a fundamental role in the modulation of mood and emotions, so alterations in these systems induced by glyphosate could favor the onset of depression or anxiety, according to observations in some studies analyzed herein [3,62,64,65].

Another effect observed on neurotransmitter processes is the alteration of synaptophysin expression after perinatal exposure to glyphosate or GBH. Specifically, an increase in synaptophysin expression was observed in the dentate gyrus and CA3 regions of the hippocampus, while its expression was downregulated in the cingulate gyrus [68]. Synaptophysin is a membrane protein present on the surface of synaptic vesicles that is expressed in most CNS neurons [107]. Glyphosate-induced changes in the expression of this protein could indicate its ability to alter neurotransmitter release and metabolism in various brain regions.

#### 3.2.3. Effects on Behavior

Exposure to glyphosate also induces important changes in rodent behavior, possibly because of alterations in neurotransmission. Early or late exposure to glyphosate or GBH has been shown to cause a decrease in locomotion, which could be associated with the changes in the dopaminergic system discussed previously [62,64,65,67,70]. Furthermore, an increase in anxiety levels and depression-like behavior of the animals was also observed [3,62,64,65]. In general, alterations in mood could arise because of the decrease in serotonin and noradrenaline levels caused by glyphosate. Likewise, cognitive functioning also seems to be affected by the action of this compound. In this regard, it was shown that exposure to GBH or glyphosate caused an impairment in learning and memory processes, which could be due to alterations in the functioning of the cholinergic system [63,65,67,74].

However, early exposure to glyphosate not only affects offspring behavior but also maternal behavior. Glyphosate and GBH treatments altered maternal licking behavior toward pups [68]. The changes induced by glyphosate in maternal behavior could favor the subsequent development of behavioral alterations in the offspring, such as the decreased social activity in adult mice observed in a study by Ait-Bali et al. [64].

#### 3.2.4. Induction of Oxidative Stress and Inflammation

Many of the most widely used pesticides worldwide exert their neurotoxic effects through oxidative stress mechanisms. Oxidative stress occurs when there is an imbalance between the production of ROS and the antioxidant capacity of the system responsible for detoxifying these reactive products. Consequently, oxidative damage of essential biomolecules, such as proteins, lipids, and DNA, occurs. The nervous system is particularly vulnerable to oxidative damage, mainly due to its low level of antioxidant activity, high oxygen requirement, and high lipid composition [108,109].

Glyphosate has been shown to induce oxidative stress immediately after administration, evidenced by an increase in LPO, which is induced by free radical action [37]. Oxidative stress induced by glyphosate exposure was also evidenced by alterations in protein and non-protein thiol concentrations [69,77]. Thiols are powerful antioxidants with the capacity to protect the organism from oxidative attack, and changes in their levels are used as indicators of the antioxidant status of organisms [110,111].

The enzymatic antioxidant defense system of living cells constitutes an adaptive mechanism in which the activities of the enzymes superoxide dismutase (SOD), catalase (CAT), and glutathione peroxidase (GPx) stand out [108,112]. Studies have shown that glyphosate can alter the activity of these enzyme systems in the nervous system of intoxicated animals. Decreases in SOD, CAT, and peroxidase activity [3,63,69], as well as increases in GPx activity and SOD expression [76] have been observed. In this regard, it is important to note that any change in the expression or activity of antioxidant enzymes, whether increased or decreased, is indicative of oxidative stress [113].

The oxidative damage induced by glyphosate was also confirmed by the inhibition of the enzymatic activity of gamma-glutamyl transferase (GGT) and glucose-6-phosphate dehydrogenase (G6PD), which are involved in the synthesis and reduction of glutathione (GSH), respectively [3,37]. GSH is one of the most efficient intrinsic antioxidants in the brain, and its decreased availability markedly promotes the production of free radicals that enhance oxidative damage [114,115]. The ability of GBH to decrease GSH content after acute exposure has been documented, although the change was not sustained over time [3,37]. This inhibitory effect was also observed for glutathione-S-transferases (GSTs), which are involved in catalyzing the conjugation of GSH to various hydrophobic and electrophilic substrates, with the aim of protecting cells from oxidative attack [116]. Therefore, it is possible that after the initial depletion of GSH, the organism undergoes various adaptive modifications to restore the levels of this antioxidant to normal concentrations in order to mitigate the neurotoxic consequences of exposition to glyphosate.

On the other hand, the main source of ROS inside cells is the mitochondria, organelles whose function is also severely altered under oxidative stress conditions. In this regard, Da Silva et al. [77] showed that in vitro exposure to GBH in astrogliomas inhibited the activity of mitochondrial respiratory chain enzymes and creatine kinase (CK), an enzyme related to energy metabolism in nervous tissue. The effect of glyphosate on mitochondrial functioning and cell survival was previously demonstrated in a study by Astiz et al. [117]. These authors demonstrated that glyphosate alone or in combination with other pesticides induced loss of mitochondrial membrane potential and reduced concentrations of cardiolipin, a phospholipid involved in the electron transport chain that is highly vulnerable to oxidative stress due to its richness in fatty acids, leading to increased LPO and neuronal death in the substantia nigra. Taken together, these results suggest that glyphosate and GBH can severely alter the functioning of mitochondria and consequently cause their elimination by mitophagy, as evidenced by the loss of mitochondrial mass observed by Da Silva et al. [77]. It would be interesting for future research to clarify whether mitochondrial dysfunction is a cause or a consequence of oxidative stress caused by glyphosate.

Neuroinflammation appears to be another process contributing to neurotoxicity induced by glyphosate. The inflammatory reaction plays a healthy role in helping the immune system cope with certain pathological conditions, but when this reaction becomes unbalanced or prolonged over time, it can damage the CNS [118]. During the inflammatory process, activation of microglia and astrocytes occurs, which release a variety of molecular signals, such as TNF-α, that contribute to the inflammatory state of the CNS [119,120]. Furthermore, during the process, activated glial cells expressing the enzyme nitric oxide synthase (NOS) can generate excessive amounts of NO, a molecule that leads to the formation of reactive nitrogen species that promote oxidative damage in the brain [121].

In this regard, the data analyzed in the present review show that exposure to glyphosate or GBH produces some proinflammatory effects measured as increases in the number and activation of microglia and astrocytes and increased expression of TNF-α in the CNS of mice [64,69], as well as increased concentrations of NO in murine PNS [78,79].

#### 3.2.5. Induction of Apoptosis and Autophagy

All of the neurotoxicity mechanisms triggered by glyphosate discussed previously could ultimately lead to neuronal death. Several studies have shown that exposure to glyphosate at any life stage reduces cell viability [37,75,76,77]. Specifically, it has been shown that both apoptosis and autophagy processes could be involved in the glyphosate-induced decrease in neuronal viability and neuronal death in rodents.

Apoptosis is a form of programmed cell death in which cells induce their own death through different pathways when subjected to certain types of stimuli. When the mitochondrial apoptotic pathway is activated, the Bax protein, which promotes neuronal death, inserts into the mitochondrial membrane and enables the release of cytochrome C into the cytoplasm, which ultimately leads to cell death [122]. In line with this, it has been observed that glyphosate treatment led to an increase in Bax protein expression while reducing levels of the anti-apoptotic protein Bcl-2 [75]. Bcl-2 promotes cell survival and inhibits the action of apoptotic proteins [123,124,125]. These data show the pro-apoptotic action of glyphosate in rat PC12 cells.

On the other hand, autophagy constitutes a mechanism for the turnover or destruction of dysfunctional or unnecessary cytoplasmic components within cells [126]. In a study by Gui et al. [75], it was observed that glyphosate treatment increased the concentration of the lipidated form of microtubule-associated protein 1A/1B light chain 3 (LC3-II), which is used as a marker for autophagosomes. The authors hypothesize that it is possible that glyphosate-induced autophagy is related to the need to remove cellular components damaged by this compound, such as mitochondria. Furthermore, these results were corroborated by the glyphosate-induced increase in the expression of the gene encoding Beclin-1, a protein that acts as a potential regulator of both [75].

### 3.3. Effects of Glyphosate in Fish

The zebrafish (*Danio rerio*) is a small freshwater teleost fish classically used as an experimental model for biomedical research and aquaculture breeding. After the mouse, the zebrafish has become one of the most widely used animal models for the study of various types of alterations in the nervous system, such as neurodegenerative and motor diseases, as well as for the study of neurotoxicity of environmental pollutants [127,128,129]. The main advantages of zebrafish, besides the high fecundity rate, rapid development, and low cost, are the organization of its nervous system, which is very similar to that of the human nervous system, and the similarity between the neurotransmitter systems [130,131].

Concerning the effects of glyphosate on fish, many of the studies included in this review describe the toxic effects of glyphosate or GBH on the nervous system of zebrafish, although the effects on other fish species have also been analyzed. Consistent with observations in for rodents, the analyzed studies show that glyphosate mainly affects nervous system development, neurotransmission, behavior, and energy metabolism, as well as producing oxidative stress and inflammation. The effects of glyphosate on fish are described in Table 4.

#### 3.3.1. Development of Nervous System

Early exposure to glyphosate or GBH causes dysregulation of genetic pathways directly involved in neuronal physiology and synaptic transmission in zebrafish, especially affecting the development of the forebrain, midbrain, and eye structure [137,148]. Dysregulation of these pathways can lead to the appearance of different alterations and malformations in the brain, such as those observed in the studies of Lanzarin et al. [140], Roy et al. [148], and Zhang et al. [151].

Likewise, it has also been shown that treatment with high doses of glyphosate during development can alter the dynamics and structure of the components of the cell cytoskeleton [135]. Specifically, exposure to glyphosate reduced the levels of acetylated α-tubulin in the polymeric fraction of zebrafish embryos, suggesting a decrease in microtubule stability.

#### 3.3.2. Effects on Behavior

The use of the zebrafish model to evaluate behavioral alterations provides a series of very relevant information for the investigation of the effects of environmental toxicants on the nervous system. Half of the studies reviewed herein reported glyphosate-induced behavioral alterations in these animals. In a study by Zhang et al. [151], an increase in locomotor activity of larvae during the day was observed, which, according to the authors, could be due to damage in the axons of primary motor neurons caused by glyphosate. In contrast, in the investigations of Forner-Piquer et al. [137] and Bridi et al. [134], early exposure to glyphosate or GBH caused a decrease in locomotion in both larvae and adults. It is possible that this apparent contrast is due to the difference in doses and exposure times used in each study. An increase in locomotion was documented by Zhang et al. [151] when exposing animals to the highest doses used (100–600 mg/L), whereas Forner-Piquer et al. [137] and Bridi et al. [134] used doses in the 0.05 to 10 mg/L range.

Another change observed after treatment with glyphosate and GBH was a decrease in aggressive behavior [134], which could be related to alterations in serotonergic neurotransmission [152]. Likewise, in a study by Faria et al. [130], glyphosate exposure caused impairment of exploratory and social behavior consistent with increased anxiety. However, the team of Lanzarin et al. [141] did not find such an association between GBH treatment and alterations in exploratory behavior and anxiety, the latter measured by alterations in hormone cortisol. Exposure to glyphosate also induced avoidance in the presence of aversive stimulus [134,141]. These results are likely related to the decreased exploratory ability observed in fish after glyphosate exposure. Finally, the occurrence of memory impairment in zebrafish after exposure to GBH was also documented, which could be related to an incorrect functioning of the cholinergic system in these animals [134].

#### 3.3.3. Effects on Neurotransmission

Several of the studies included herein investigated glyphosate-induced changes in AChE activity in various species, and the data reflect a heterogeneity of effects. Sobjak et al. [150] observed that exposure of eggs and larvae of silver catfish (*Rhamdia quelen*) to glyphosate caused an unexpected increase in AChE activity at 48 h after exposure, followed by a decline at 96 h. These results suggest that exposure to glyphosate during early developmental stages may cause an early induction of AChE activity and a subsequent difficulty in maintaining elevated AChE activity at later times.

On the other hand, regarding the effects of glyphosate on AChE in young and adult fish, several studies support its inhibitory potential against the activity of this enzyme [132,133,145]. This enzymatic inhibition would affect the anterior and middle regions of the fish body and seems to depend on the concentration and time of year, being more potent in autumn [132,145]. In contrast, in a study by Teixeira et al. [136], an increase in brain AChE activity was observed, whereas Jin et al. [138] and Lopes et al. [144] did not detect any alteration in its activity level. The considerable variability of these results is probably due to the wide diversity of species and doses of glyphosate used in each of the investigations.

Another of the changes observed is an alteration of the balance between glutamine, glutamate, and the amino acid gamma-aminobutyric acid (GABA), which are fundamental components of brain metabolism and function. Glutamine is produced from glutamate reuptake by astrocytes, and although it does not fulfill a neurotransmitter function, it is the main precursor for glutamate and GABA synthesis [153]. In this regard, Li et al. [142] observed that GBH exposure reduced both GABA and glutamate, resulting in an elevation in brain glutamine levels and the subsequent appearance of various behavioral alterations.

Furthermore, Faria et al. [130] documented an increase in the content of dopamine and its acidic metabolites (3,4-dihydroxyphenylacetic acid, DOPAC, and homovanillic acid, HVA) and in dopamine turnover in the forebrain. However, no variations were found in the levels of 3-methoxytyramine, an extracellular metabolite of dopamine. Therefore, because DOPAC is an intracellular metabolite and HVA reflects both intra- and extraneuronal metabolism, the changes observed after glyphosate exposure suggest that it could increase the intraneuronal metabolism of dopamine but without inducing increased release of the neurotransmitter. Glyphosate also downregulated the expression of several genes involved in dopamine synthesis, degradation, and transport [130]. In this same study, the authors also reported that glyphosate exposure increased serotonin levels in the forebrain, which could influence the mood and behavior of the animals.

The alterations in neurotransmission could explain the behavioral alterations observed by Sánchez et al. [149] in one-sided livebearer (*Jenysia multidentate*). In this study, exposure to 0.5 mg/L GBH for 96 h was associated with a decrease in spatial exploration and swimming performance, as well as impaired long-term memory consolidation. Furthermore, these authors also observed a negative effect of GBH on social interaction and sexual behavior of fish.

#### 3.3.4. Induction of Oxidative Stress and Inflammation

As previously discussed, many of the most widely used pesticides exert their neurotoxicity by altering the balance between ROS production and the ability of the organism’s antioxidant system to neutralize them. In this regard, Faria et al. [130], Pereira et al. [147], and Sobjak et al. [150] have documented the ability of glyphosate and GBH to increase ROS concentrations in the nervous system of fish, as well as the LPO caused by them. The exception is the work of Lopes et al. [144], wherein a decrease in LPO levels was observed in zebrafish after exposure to glyphosate. Although the team of Teixeira et al. [136] found no changes in LPO levels after treatment of pintado da Amazônia fish with sublethal doses of GBH, they did observe an increase in carbonyl protein content. These data suggest the existence of an alteration in normal protein metabolism that can be used as a marker of protein oxidative damage.

Mitochondria are cellular organelles that contribute considerably to ROS production and thereby to oxidative stress. In a study by Pereira et al. [147], in vivo exposure to GBH induced mitochondrial dysfunction in brain cells after 7 days of treatment, evidenced by a decrease in the activity of the nicotinamide adenine dinucleotide (NADH) dehydrogenase and cytochrome C, enzymes of mitochondrial complexes I and IV, respectively. These alterations were also associated with changes in the transcription of genes encoding electron transfer proteins in the mitochondrial respiratory chain and a subsequent hyperpolarization of the mitochondrial membrane potential. In this regard, when the mitochondrial membrane potential is high, the mitochondrial respiratory chain becomes a producer of ROS, which poses a serious risk to cell integrity [154].

GBH exposure also caused a decrease in the levels of choline; its derivative, phosphocholine; and its metabolite, betaine, in the brain. Choline constitutes an essential component of different phospholipids of lipid bilayers and is necessary for the maintenance of structural integrity of cell membranes [155]. Therefore, because cell membrane phospholipids are the main target of oxidative attack, the decrease in levels of choline and its derivatives after glyphosate exposure could be indicative of an accelerated use of these components to repair cell membranes impaired by free radicals.

Another marker of oxidative stress induced by exposure to neurotoxic agents is the alteration in the activity of enzymes responsible for scavenging free radicals and restoring the normal antioxidant status of nerve tissue. It has been documented that exposure to glyphosate induced an increase in the activity of the antioxidant enzymes CAT, SOD, and glutathione reductase (GR), with a concomitant decrease in GSH reserves, which could reflect an attempt by the cells to neutralize the excessive levels of ROS caused by glyphosate [130]. However, Sobjak et al. [150] found no changes in CAT and GST activity in the brain of silver catfish after exposure to glyphosate.

Glyphosate exposure also appears to induce an inflammatory reaction in the nervous system of zebrafish. This was demonstrated by a study by Forner-Piquer et al. [137], who administered concentrations of 0.05 and 10.000 μg/L glyphosate and detected the presence of amoeboid cells, suggesting microglial activation and therefore activation of the inflammatory process. In addition, transcriptomic analysis revealed dysregulation of genetic pathways involved in inflammation. In contrast, Li et al. [142] demonstrated that exposure of goldfish (*Carassius auratus*) to 0.22, 0.44, or 0.88 mmol/L of GBH induced a decrease in the levels of myoinositol, a compound that is mainly present in glial cells and is therefore considered a glial marker [156].

#### 3.3.5. Effects on Energy Metabolism

The effect of glyphosate on brain energy metabolism is an aspect not addressed in studies in other species analyzed so far, although the available evidence supports the potential of pesticides to interfere with energy production [157]. Glucose is the main source of energy in the brain, and its catabolism via glycolysis results in pyruvate, which is subsequently transferred to mitochondria to produce adenosine triphosphate (ATP) via the tricarboxylic acid (TCA) cycle and the mitochondrial respiratory chain [158]. GBH exposure was shown to deregulate glucose metabolism in the goldfish brain, which was evidenced by decreased succinate and citrate concentrations [143]. Because both are intermediates of the TCA cycle, the reduction in their levels suggests that glyphosate may block the TCA cycle and thereby generate an insufficient amount of ATP.

To survive this energy crisis, the body can use additional resources of energy production. For example, creatine and phosphocreatine are important energy metabolites because phosphocreatine hydrolysis donates a phosphate group to adenosine diphosphate (ADP) to synthesize ATP [159]. In this regard, a decrease in creatine and phosphocreatine levels was observed in the brain of fish dosed with GBH, suggesting the activation of the phosphocreatine reaction with ADP in order to supply the need of ATP [142]. Likewise, Li et al. [143] observed increased levels of ketone 3-hydroxybutyrate in the brain tissue of GBH-treated goldfish. In the brain, ketones are the main alternative fuel to glucose, so they could be used to meet the insufficient energy supply caused by glyphosate exposure [160].

When glucose is unable to maintain brain homeostasis, other substrates, such as acetate, can also be used as metabolic precursors [161]. In line with this, Li et al. [142] found that GBH exposure induced a considerable reduction in brain levels of the amino acid N-acetyl-L-aspartate (NAA), the second most abundant amino acid in the CNS, which can be degraded to aspartate for energy production [162].

On the other hand, adenosine monophosphate (AMP)-activated kinase is the main sensor and regulator of cellular energy homeostasis. Under low-energy conditions, this kinase is activated and inhibits ATP-using (anabolic) processes while stimulating ATP-producing (catabolic) processes [163]. Therefore, the reduction in AMP levels in brain tissues of fish exposed to GBH described by Li et al. [143] may affect the AMP-activated kinase pathway and thus alter downstream metabolic processes. In contrast to these results, in a study by Menéndez-Helman et al. [146], GBH treatment did not produce alterations in adenylate energy charge (AEC), a measure reflecting the amount of energy available from the adenylate pool (ATP, ADP, and AMP).

Overall, the changes at the molecular level induced by glyphosate discussed previously can lead to the manifestation of important alterations both in structure and in brain function. This is clearly reflected in the destruction of the microscopic structure of the brain or in the appearance of abnormal and variable brain activity in the midbrain of fish exposed to glyphosate alone or in formulation [137,141]. Nevertheless, when considering these studies, the heterogeneity of the obtained results is evident, which, as mentioned above, may be due to the use of different species, as well as to treatment with doses and exposure periods that differ considerably. This makes it difficult to draw rigorous conclusions and demonstrates the need for more research in this area.

### 3.4. Effects of Glyphosate on Invertebrates

The number of recent studies on the effects of glyphosate on the nervous system of invertebrates is limited. Two studies were identified with the worm *Caenorhabditis elegans* describing that GBH also produces neurotoxicity in this species, which was evidenced by altered neuronal development, mitochondrial damage, oxidative stress, and in behavioral patterns.

On the one hand, it has been shown that GBH treatment affected cell development in worms, which was evidenced by a decrease in the number and size of dopaminergic neurons from the fourth larval stage onwards [164]. These authors also observed that GBH caused a marked increase in superoxide levels during the fourth larval stage [164] and hydrogen peroxide in treated worms [165]. As has been observed in rodents, GBH exposure also affects mitochondrial function in these animals. In addition, Burchfield et al. [165] showed that glyphosate induced inhibition of mitochondrial complex II, and consequently, a decrease in ATP levels occurred.

## 4. Discussion

### 4.1. Overview of the Main Mechanisms of Action of Glyphosate on the Nervous System

A wide variety of studies show the correlation between pesticide exposure and the development of various types of diseases. Organophosphate exposure has been reported to be associated with various human conditions, such as mood disorders, attention deficit hyperactivity disorder, cancer, kidney damage, and autism, among others [166,167,168,169]. Furthermore, it has been postulated that pesticides may be the main environmental factor associated with the etiology of neurodegenerative diseases, such as Alzheimer’s and Parkinson’s disease [59,170].

Regarding glyphosate, some previous studies have reported DNA damage in human and rodent cells after exposure to this compound and GBH [171,172,173]. The teratogenic action of GBH in vertebrates was also described [174]. Likewise, human clinical reports on the effects of intoxication with glyphosate formulations have described harmful effects on the nervous system, including parkinsonism [175].

The data analyzed in the present review show that exposure to glyphosate or GBH generates various types of neurotoxic effects in all the species studied (Figure 3). The study of glyphosate effects in humans is based on the observation of clinical signs and symptoms in people accidentally or intentionally exposed to the pesticide or from in vitro studies with human cell lines. Although these in vitro studies are quite heterogeneous, in general, the results corroborate those described for the other species studied. Some of the findings of these investigations corroborate the ability of glyphosate to cross the human BBB and produce neurotoxic effects in the CNS. This ability of glyphosate to cross both the placental barrier and the BBB in humans was also observed in other previous studies that detected this compound in the brain and cerebrospinal fluid of individuals who had been exposed to GBH [176,177,178].

Nevertheless, it appears that in adult humans, glyphosate does not produce toxic effects immediately after exposure but takes one or two days to do so. It is possible that this delay is due, at least in part, to the fact that the pesticide takes time to alter the integrity of the BBB, cross it, and subsequently distribute itself in the CNS. Clinical observations also show that the adverse effects caused by glyphosate disappear as the concentration of the compound in the body decreases, although there are certain sequelae that seem to be maintained over time, such as memory disturbances.

Figure 4 summarizes the main mechanisms by which glyphosate produces its toxic effects on the nervous system. However, the existence of important discrepancies between the results obtained in studies carried out with humans, rodents, and fish must be considered. The data analyzed suggest that glyphosate induces the entry of Na^+^ and Ca^2+^ from the extracellular medium through different mechanisms. Glyphosate causes an overstimulation of NMDAR, which leads to an influx of these ions and membrane depolarization. On the other hand, the inhibition of AChE by glyphosate produces an increase in acetylcholine levels and the stimulation of its receptors, including the α7 and α4β2 receptors. The opening of these channels also induces an Na^+^ and Ca^2+^ influx. Furthermore, although α4β2 receptors are rapidly desensitized, α7 receptors can remain open for longer in the presence of agonists, further increasing Ca^2+^ influx [179]. It is possible that the sustained activity of these receptors induced by chronic exposure to glyphosate eventually leads to a decrease in their expression. Taken together, depolarization resulting from activation of glutamatergic and cholinergic receptors can cause VDCCs to open, allowing more Ca^2+^ to enter the cell. Some studies have shown that because of excess glutamate and acetylcholine, their metabotropic receptors are also activated [37,180,181]. The mGluR1 and mGluR5 receptors and mAChRs 1, 3, and 5 act on the same signaling pathway by activating associated G proteins, which in turn activate phospholipase C and the production of inositol triphosphate (IP3). The resulting IP3 diffuses through the cytoplasm and binds to specific receptors in the endoplasmic reticulum (ER), causing the release of Ca^2+^ to the cytosol [182].

Similarly, exposure of human cells to glyphosate has been shown to induce an increase in the expression of Wnt-5a mRNA, although this increase was not observed in rodents. Wnt-5a can interact with the Frizzled receptor, the activation of which also leads to an increase in IP3 levels and the consequential release of Ca^2+^ to the cytoplasm [183]. Likewise, Ca^2+^ ions inside the cell can bind to ryanodine receptors, Ca^2+^ channels present in the ER, the opening of which allows for the exit of this cation into the cytoplasm [182]. Ca^2+^ is an intracellular messenger involved in multiple signaling pathways, the concentrations of which are strictly controlled. Thus, this increase causes numerous changes in the cell. One such change is its binding to calmodulin, which leads to the activation of neuronal nitric oxide synthase (nNOS), which contributes to oxidative stress through the release of large amounts of NO [184].

As has been amply demonstrated, glyphosate produces a marked increase in intracellular ROS concentrations, which induce the activation of a series of mechanisms to neutralize oxidative stress conditions. Among these mechanisms is the activity of endogenous antioxidants, both enzymatic (SOD, CAT, GPx, peroxidase (PRX), GST, GGT, and G6PD) and non-enzymatic (GSH and thiols). However, these mechanisms appear to be insufficient to counteract the large amount of free radicals resulting from exposure to the pesticide. ROS act on the membranes of organelles, proteins, and DNA, causing their oxidation. In this situation, the cell can adopt various strategies to repair its components, such as increasing the activity of alkaline phosphatase to repair its DNA.

On the other hand, glyphosate has been shown to induce neuroinflammation characterized by the activation of both microglia and astrocytes. Because of their activation, these cells release large amounts of inflammatory cytokines, such as TNF-α and IL-6. In addition, exposure to glyphosate could also promote the release of the S100B protein by astrocytes, reaching its maximum levels one day after intoxication. At micromolar concentrations, the S100B protein can act on the receptor for advanced glycation end products (RAGE) at the neuronal membrane, favoring the overproduction of ROS [185]. Furthermore, this protein also binds to RAGEs present in the glial membrane, where it amplifies the inflammatory response [186]. In line with this, both the S100B protein and some inflammatory cytokines can trigger inducible nitric oxide synthase (iNOS) activation in astrocytes [187]. Consequently, these glial cells produce large amounts of NO, which further enhances oxidative stress.

To eliminate the organelles damaged by the action of ROS and obtain energy, the cells can induce the process of autophagy. Thus, the increase in Ca^2+^ and ROS concentrations caused by glyphosate has been related to increased activation of ERK, which can activate the Beclin 1 protein and induce autophagy [188]. In addition, this natural recycling process can also be induced by other mechanisms observed in studies with glyphosate, such as the binding of the Wnt-3a ligand to the Frizzled receptor or through the deficient production of ATP in mitochondria [189].

Although autophagy favors cell survival in stressful situations or under nutrient deprivation, excessive induction of this mechanism can be destructive to the cell and lead to apoptosis [190]. The conditions of oxidative stress induced by glyphosate can trigger the intrinsic or mitochondrial apoptotic pathway because the ability of these organelles to produce energy and maintain Ca^2+^ homeostasis is severely affected. Therefore, the increase in the intracellular Ca^2+^ also leads to a marked increase in its levels in the mitochondria. Consequently, activation of the Bax protein occurs. This activated protein forms pores in the mitochondrial membrane and allows for the release of cytochrome C, which activates the caspases responsible for orchestrating apoptosis [191].

Exposure to glyphosate can also activate the extrinsic apoptotic pathway. As previously mentioned, glial activation by the pesticide causes the release of inflammatory cytokines, such as TNF-α, which can act as a ligand for the death receptors at the membrane. As a result of the activation of these receptors, the activation of caspase 8 occurs, which clears other caspases and finally causes neuronal death by apoptosis [192].

These findings also suggest that when exposure occurs early in life, the pesticide can induce serious disturbances in the development of the nervous system, possibly by deregulating some of the signaling pathways involved in this process. On the other hand, a common aspect that can be drawn from in vivo and in vitro studies with rodents and humans is that both in early age and in adulthood, the hippocampus is especially vulnerable to the action of glyphosate. As a result of this damage, exposure to different concentrations of glyphosate has usually been associated with severe memory impairment that is sometimes irreversible.

### 4.2. Relationship between Glyphosate Doses and Neurotoxic Effects in Rodents and Humans

The findings reported in this review coincide in pointing out the neurotoxic potential of glyphosate in the different animal species studied. Although many of the studies used doses higher than those commonly found in the environment, the main objective of these studies was the evaluation of the neurotoxic mechanisms of action of the pesticide on the nervous system. Therefore, to assess the real toxic risk that exposure to glyphosate may pose in animals, it is necessary to consider environmental concentrations of the pesticide.

The first point to consider is that there are several discrepancies with respect to the values established as NOAEL. Several previous investigations have established a NOAEL for glyphosate at considerably high values, such as 1000 mg/kg/day, which established for maternal and developmental toxicity in rats [14]. However, these guideline values were established considering carcinogenicity and toxicity in organs other than the brain. Therefore, a considerable number of regulatory toxicity studies found no evidence that high doses of glyphosate caused neurotoxic effects in rodents. This has led to some subsequent studies to establish a NOAEL for neurotoxicity in rats ranging from 2000 to 20.000 mg/kg/day of glyphosate [193]. Finally, more recent research established a more moderate NOAEL value of 617 mg/kg/day [34].

Most of studies of rodents analyzed in this review used doses of glyphosate or GBH that did not exceed the current NOAEL and ranged from 5 mg/kg [68] to 800 mg/kg [13]. These doses are not representative of human environmental exposures, which are in the range of μg/kg/day. However, the data provided by these studies suggest that even oral treatment with the lowest used dose of glyphosate or GBH (5 mg/kg) was associated with alterations in behavior and neuroplasticity in rats [68]. Likewise, the dose of glyphosate most used in the studies with rodents was 50 mg/kg, which is also much lower than the established NOAEL, and it was associated with important alterations in the structure and functioning of the CNS. Therefore, although it appears that more conservative guideline values for the neurotoxicity of glyphosate and GBH have been adopted in recent years, the data discussed herein suggest that these values may still be too high. The results analyzed in this review suggest that the ability of glyphosate, alone or in formulation, to produce neurotoxic effects in rodents has been underestimated.

The current evidence analyzed in this review raises concern and indicates the need for more research, mainly in rodents. This is because rodent studies provide essential information for the establishment of reference values for humans. In line with this, the reference dose of glyphosate established by the EPA (USA) is 1.75 mg/kg/day, whereas the JMPR/WHO is more restrictive and limits the maximum dose to 1 mg/kg/day [194]. The EPA has estimated that the exposure of the general population to glyphosate through food and water is 0.088 mg/kg/day (range 0.058–0.230 mg/kg/day) [195], some studies have indicated that glyphosate is habitually found in urine in concentrations corresponding to a daily intake of approximately 0.1–0.3 μg/kg/day [28]. These data indicate that the general population is exposed to glyphosate levels lower than the reference dose and the acceptable daily intakes proposed by several regulatory agencies, not representing an immediate risk to health. However, under certain circumstances, the established reference values can be clearly exceeded, as in the case of some accidental or intentional exposures. Thus, the most serious neurotoxic effects reported have been observed after suicide attempts, with exposures to doses of glyphosate much higher than those normally found in the environment.

Studies on occupational toxicity in rural populations are of particular relevance. In such studies, glyphosate concentrations commonly used in agricultural practice or those found in residential environments close to growing areas were considered. These studies reveal a possible relationship between exposure to glyphosate during prenatal and childhood and an increased risk of developing diseases such as autism spectrum disorder [47]. Likewise, peripheral neuropathy and memory impairment have also been reported in farmers occupationally exposed to the pesticide [45,196]. These data suggest that pesticide concentrations found in agricultural settings could represent a health risk to children and adults.

### 4.3. Exposure Levels and Neurotoxic Effects in Fish

When GBH is applied to crops, it can remain in the soil until it is degraded by microorganisms or mobilized by wind, rain, or irrigation. These factors increase infiltration and surface runoff, favoring the arrival of glyphosate in aquatic environments [146]. Varied concentrations of glyphosate have been reported in aquatic environments, such as concentrations of 0.1–1.48 mg/L detected in waters of Argentina and Brazil, or the maximum concentration of 0.65 mg/L detected in waters in Europe [149,197]. During the glyphosate use season, its concentration reaches 10 mg/L in Taihu Lake (China) [48].

Therefore, Roundup^®^ concentrations below 10 mg/L can be considered environmentally realistic considering that with current application rates, a bare water body can have a maximum glyphosate concentration of 3.7 mg/L, which corresponds to 9 mg/L of Roundup^®^ [198]. Almost all the fish studies analyzed in this review used doses that ranged from 0.064 to 10 mg/L of GBH or glyphosate. Exposure to these environmentally relevant doses of glyphosate was associated with a wide range of neurotoxic effects, including alterations in the development of the nervous system and the appearance of alterations in behavior. In addition, although, given the physicochemical characteristics of glyphosate, it is not expected that it can bioaccumulate in the tissues of animals, the available evidence supports that this compound can accumulate in the tissues of some aquatic species, which could result in an increase in its neurotoxicity [199,200,201,202].

Although elevated concentrations of glyphosate have been detected in some aquatic environments, some countries have adopted more restrictive limits. Thus, the maximum level of glyphosate accepted in the United States is 0.7 mg/L, whereas in Europe, the maximum accepted for a standard environmental quality is 0.028 mg/L [43,203]. In line with this, Bridi et al. [134] showed that exposure of zebrafish to glyphosate or GBH (0.01–0.5 mg/L) caused alterations in locomotion, aversive and aggressive behavior, as well as memory impairment. In a study by Khan et al. [139], treatment with 0.02–0.1 mg/L of glyphosate also altered the behavior in European carp (*Cyprinus carpio)*. Similarly, Faria et al. [130] found that the exposure of zebrafish to concentrations of 0.3 or 3 μg/L of glyphosate for two weeks caused a deterioration in exploratory and social behavior, as well as an increase in anxiety. Likewise, the team of Sánchez et al. [149] observed that treatment of one-sided livebearer (*Jenynsia multidentate*) with 0.5 mg/L GBH reduced social interaction, spatial exploration, swimming performance, and long-term memory consolidation and even caused severe alterations in sexual behavior.

These results suggest that exposure to glyphosate or GBH, although in doses that are within legally accepted limits, can result in neurotoxicity in fish and alter behaviors essential for the survival of these species. Ultimately, this could have important ecological consequences, especially in agricultural areas where agrochemicals are applied regularly. Therefore, these results suggest the need to adopt more conservative contamination limits that consider the neurotoxicity induced by glyphosate.

## 5. Limitations

This systematic review is not free of limitations, which should be considered when generalizing the results. First, most of the studies analyzed used commercial formulations of glyphosate, which are mixtures of glyphosate (active ingredient) with other adjuvants. Thus, it is not possible to determine precisely which component(s) of the formulation was (or were) responsible for the observed neurotoxic effects. In addition, the results of various studies support the greater toxicity of commercial glyphosate formulations compared to glyphosate administered alone [147,204,205,206]. For this reason, the results analyzed herein cannot be attributed exclusively to glyphosate, as they could have been caused by other components of the formulation or even by possible synergy between these components and glyphosate [65,207,208,209].

On the other hand, most research has studied the effects of higher doses of glyphosate than the concentrations to which the general population is routinely exposed. Thus, it is possible that exposure to environmental concentrations of glyphosate does not result in the wide range of neurotoxic effects documented here. Furthermore, when analyzing many studies, a great heterogeneity in the results obtained is observed; the various discrepancies identified when comparing the studies could be due to differences in their experimental conditions, such as the route of exposure, the compound used (glyphosate, AMPA, or GBH), the total duration of treatment, and the time at which the evaluations were performed since the end of the treatment.

## 6. Conclusions

The information summarized in the present review indicates that exposure to glyphosate, AMPA, or GBH could induce several toxic effects on the nervous system of all species studied. Exposure to glyphosate during the early stages of life can severely affect normal cell development by deregulating some of the signaling pathways involved in this process, leading to alterations in differentiation, neuronal growth, migration, and myelination. Glyphosate also seems to exert a significant toxic effect on neurotransmission, with the glutamatergic system being one of the most affected systems. Glyphosate was found to increase glutamate release and decreased its reuptake, in addition to activating NMDAR and L-VDCC, thus increasing the influx of Ca^2+^ into neurons. Likewise, the results analyzed herein reflect the capacity of glyphosate to induce oxidative stress, neuroinflammation, and mitochondrial dysfunction, processes that lead to neuronal death by autophagia, necrosis, or apoptosis, as well as the appearance of behavioral and motor disorders. Although there are important discrepancies between the findings analyzed in this review, it is unequivocal that exposure to glyphosate, alone or in commercial formulations, can produce important alterations in the structure and function of the nervous system of humans, rodents, fish, and invertebrate animals.

## Figures and Tables

**Figure 1 ijms-23-04605-f001:**
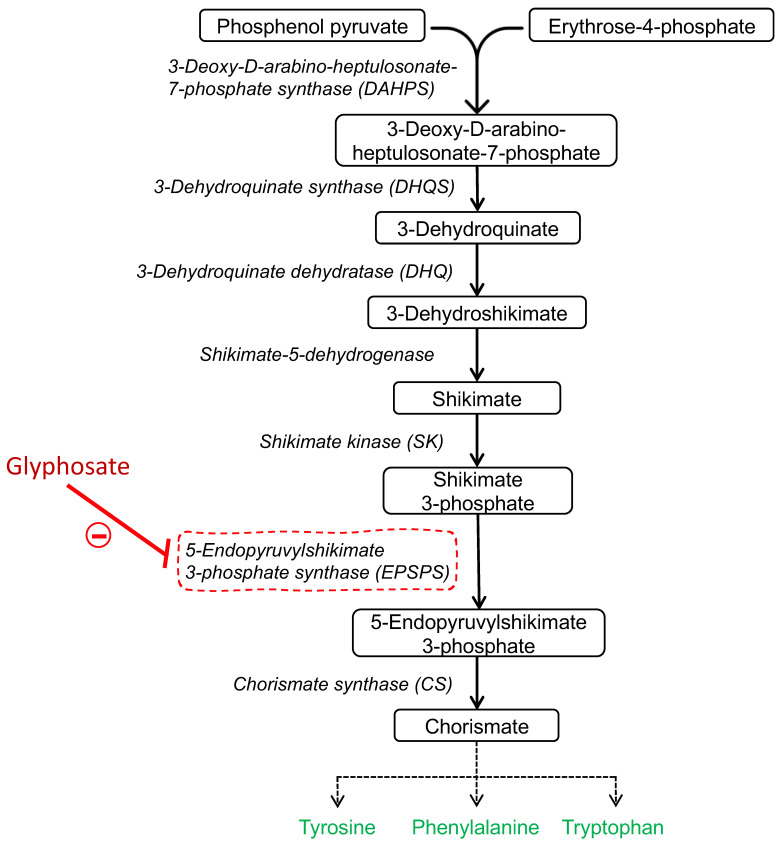
Glyphosate inhibits the enzymatic activity of the 5-endopyruvylshikimate 3-phosphate synthase (EPSPS) in the shikimic acid pathway, preventing the synthesis of the aromatic amino acids tyrosine, phenylalanine, and tryptophan.

**Figure 2 ijms-23-04605-f002:**
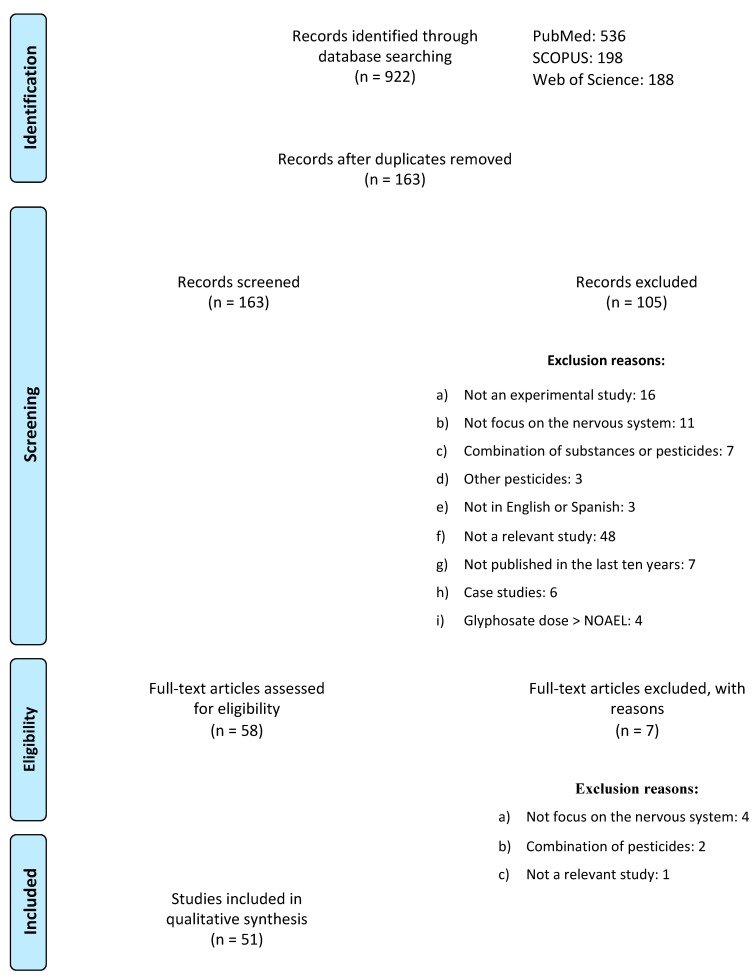
Flow diagram of the systematic search process.

**Figure 3 ijms-23-04605-f003:**
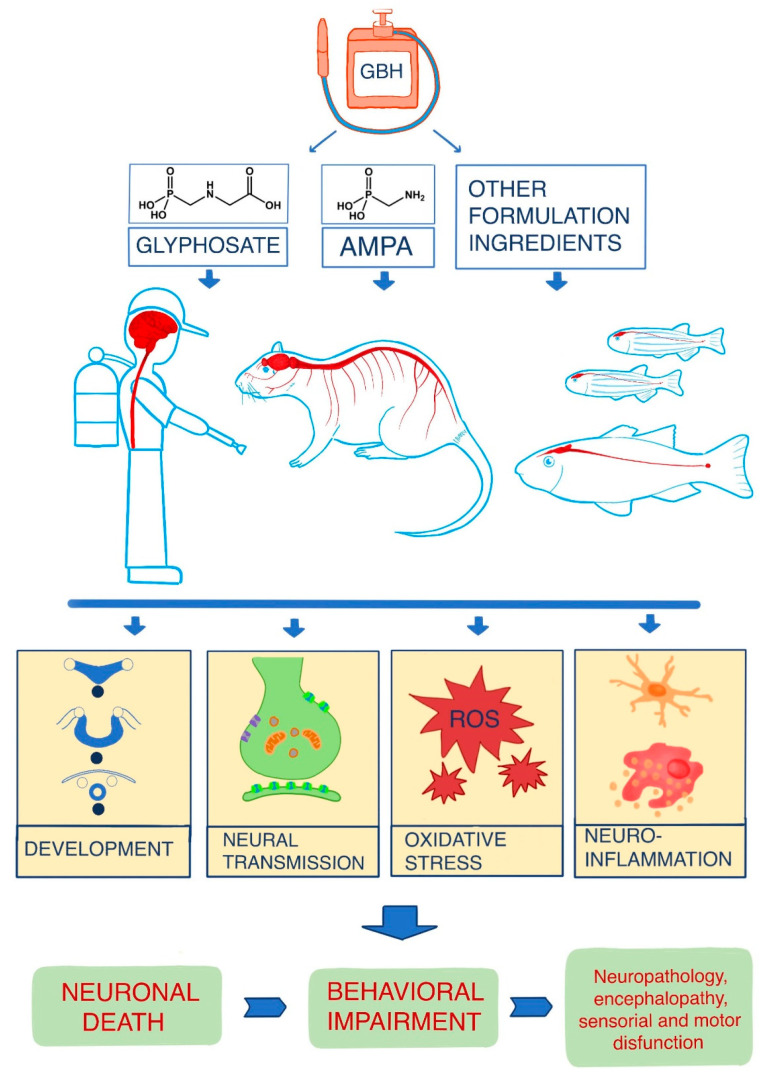
Exposition to glyphosate; its main metabolite, AMPA (aminomethylphosphonic acid); or its commercial formulations induces neurotoxic effects in all studied species. The main modes of action include changes in the development of the nervous system and in the neurotransmission systems, oxidative stress, neuroinflammation, processes that lead to neuronal death, and the appearance of behavioral changes. Changes in the structure and function of neurons lead to the development of neuropathology, encephalopathology, and sensory and motor dysfunctions.

**Figure 4 ijms-23-04605-f004:**
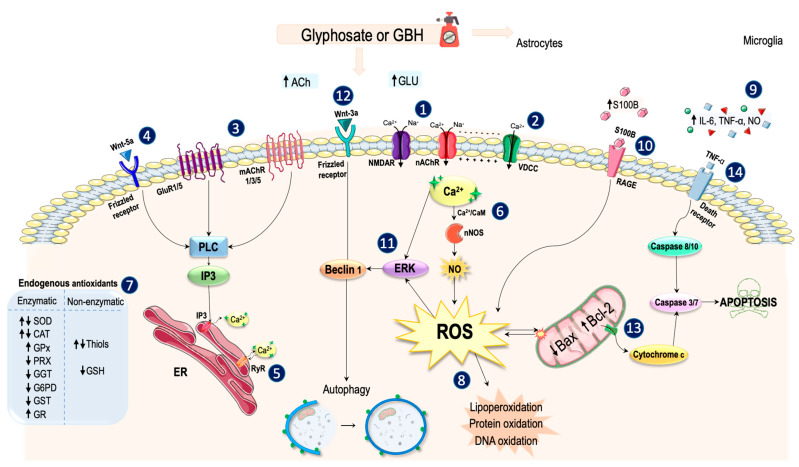
Possible mechanism of action of glyphosate or GBH in the nervous system. The presence of glyphosate induces several changes, including (1) opening of nAChRs and NMDA receptors, as well as entry of Na^+^ and Ca^2+^ into the cell due to increased levels of ACh and GLU and/or the direct binding of glyphosate to the cavities of NMDAR; (2) opening of the VDCCs by cellular depolarization and entry of Ca^2+^; (3) activation of the metabotropic GLU and ACh receptors, which stimulate the PLC to generate IP3, which causes the release of Ca^2+^ from inside the ER; (4) increase in the levels of Wnt-5a, which binds to Frizzled receptors and triggers the generation of IP3, with the consequential release of Ca^2+^ from the interior of the ER; (5) Ca^2+^ binding to ryanodine receptors and Ca^2+^ release from inside the ER; (6) binding of Ca^2+^ to calmodulin and activation of nNOs, which releases NO; (7) modification of the activity and/or concentrations of endogenous antioxidants; (8) excessive levels of ROS, leading to oxidation of lipids, proteins, and DNA; (9) activation of glial cells, which release inflammatory cytokines and NO; (10) release of S100B protein, which binds to neuronal RAGEs and increases ROS overproduction; (11) activation of ERK due to excessive levels of Ca^2+^ and ROS, which activates Beclin 1 and induces autophagy; (12) increased levels of Wnt-3a, which binds to Frizzled receptors and induces autophagy; (13) mitochondrial dysfunction, leading to activation of the intrinsic apoptosis pathway; and (14) binding of the ligand TNF-α to the death receptor, activating the extrinsic apoptosis pathway. Parts of the figure were created using templates from Servier Medical Art, which are licensed under a Creative Commons Attribution 3.0 Unported License (http://smart.servier.com/ accessed on 17 February 2022). Abbreviations: GBH: glyphosate-based herbicide; GLU: glutamate; nAChR: nicotinic acetylcholine receptor; NMDAR: N-methyl-D-aspartate receptor; VDCC: voltage-dependent calcium channel; PLC: phospholipase C; IP3: inositol trisphosphate; ER: endoplasmic reticulum; CaM: calmodulin; nNOS: neuronal nitric oxide synthase; NO: nitric oxide; SOD: superoxide dismutase; CAT: catalase; GPx: glutathione peroxidase; PRX: peroxidase; GGT: gamma-glutamyl transferase; G6PD: glucose-6-phosphate dehydrogenase; GST: glutathione S-transferase; GR: glutathione reductase; GSH: glutathione; ROS: reactive oxygen species; IL-6: interleukin-6; TNF-α: tumor necrosis factor alpha; S100B: S100 calcium-binding protein B; RAGE: receptor for advanced glycation end products; ERK: extracellular signal-regulated kinase; ↑, increase; ↓, decrease.

**Table 1 ijms-23-04605-t001:** Studies on the effects of glyphosate and/or its commercial formulations in humans.

Type of Study	Toxic Agent	Exposure Mode/Objetives	Results	Reference
Transversal study	GBH	Occupational exposure	- Positive association between GBH exposure and visual memory impairment	[45]
Prospective cohort study	GBH	Not specified	- ↑ in S100B protein levels in patients with neurological complications - S100B protein was a predictor of neurological complications in GLY-poisoned patients	[46]
Population-based case-control study	GBH	Occupational exposure	- Prenatal and infant exposure increases the risk of autism spectrum disorder - Exposure during childhood appears to increase the risk of developing more severely impaired phenotypes with comorbid intellectual disability	[47]
Cohort study	GBH	Occupational exposure	- GBHs were associated with lower toxicity to farmers’ health compared to other non-GBHs	[48]
Cohort study	GBH	Occupational exposure	- No relationship was found between GBH use and peripheral nerve conduction abnormalities in farmers	[49]
In vitro SH-SY5Y cell line	GLY alone or mixed with other formulants: 5.33 to 3.200 μg/mL for 24 h	Investigate whether GBH toxicity is related to formulants	- Inhibition of cell proliferation when GLY was administered with other formulants but not when it was administered alone	[50]
In vitro IMR90-c4 iPSCs line	GLY, AMPA: 0.1 to 1000 μM for 24 or 48 h	Investigate the effect of GLY on the BBB in vitro and compare it with that of AMPA and glycine	- GLY and its metabolite altered the integrity of the BBB - GLY can be released through the BBB - High doses of GLY and AMPA altered glucose uptake by microvascular endothelial cells in the brain and metabolic activity of neurons	[51]
In vitro SH-SY5Y cell line	GLY, AMPA: 0.1 to 20 mM for 48 h	Investigate the effects of GLY and AMPA on oxidative stress, neurodevelopment, and cell death.	- ↓ cell viability and increased leakage of LDH - ↑ production of MDA, NO, and ROS - ↑ caspase-3/7 activity - GLY ↑ the levels of mRNA-Wnt3a, -Wnt5a, and -Wnt7a - GLY positively regulated IL-6 and TNF-α genes - GLY ↑ the expression of CAMK2A and CAMK2B mRNA - GLY and AMPA downregulated the expression of the *TUBB3* and *GAP43* genes - GLY and AMPA altered the gene expression of cell death pathways	[52]

Abbreviations: GBH, glyphosate-based herbicide; ↑, increase; S100B, S100 calcium-binding protein B; GLY, glyphosate; iPSCs, induced pluripotent stem cells; AMPA, aminomethylphosphonic acid; BBB, blood–brain barrier; ↓, decrease; LDH, lactate dehydrogenase; MDA, malondialdehyde; NO, nitric oxide; ROS, reactive oxygen species; IL-6, interleukin-6; TNF-α, tumor necrosis factor alpha; CAMK2, Ca^2+/^calmodulin-dependent protein kinase 2.

**Table 2 ijms-23-04605-t002:** In vivo effects of glyphosate and/or its commercial formulations in rodents.

Species	Dose and Exposition	Time Exposition	Objectives	Results	Reference
Swiss mice	Roundup^®^: 250 or 500 mg/kg/day orally	Subchronic exposition: 6 weeks Chronic exposition: 12 weeks	Assess the effects of acute or repeated GBH exposure on the developing brain of young and adult mice	Chronic/subchronic exposure: - ↓ locomotion - ↑ anxiety, depressive behavior - ↓ 5-HT immunoreactivity Chronic exposure: ↓ TH immunoreactivity	[62]
Swiss mice	Roundup^®^: 250 or 500 mg/kg/day orally	Subchronic exposition: 6 weeks Chronic exposition: 12 weeks	Evaluate the effects of GBH on learning and memory functions, AChE activity, and oxidation/antioxidation homeostasis	Chronic/subchronic exposure: - Impaired recognition and retention memory - ↓ AChE activity - ↓ SOD and peroxidase activity Chronic exposure caused impairment in working memory	[63]
Swiss mice	Roundup^®^: 250 or 500 mg/kg/day orally	From GD0 to PND21	Evaluate the behavioral (PND5-PND25) and biochemical (PND60) effects of gestational and lactational exposure to GBH on offspring	- Delayed sensorimotor development from PND5 to PND25 - ↓ locomotion, anxiety, ↓ sociability, cognitive impairment - ↓ number of TH^+^ cells - ↓ AChE activity - Activation of microglia and astrocytes. ↑ TNF-α expression and ↓ BDNF. ↑ TrkB levels	[64]
CF-1 mice	Glifloglex^®^: 50 mg/kg/day intranasally	Three days a week for four weeks	Assess the neurobehavioral effects of repeated intranasal administration of a GBH	- ↓ locomotion - ↑ anxiety - Impaired recognition memory	[65]
Wistar rats	Roundup^®^: 70 mg/kg/day orally	Chronic exposition: from GD5 to PND15. Acute exposition: 30 min in vitro	Determine the neurotoxic effects of GBH on the hippocampal function of immature rats after chronic exposure (pregnancy and lactation) and after acute in vitro exposition.	Acute in vitro exposition: - ↑ glutamate release, activation of NMDAR and L-VDCC, and ↑ Ca^2+^ influx - CaMKII and ERK activation - LPO, ↓ GGT and G6PD activity, ↓ GSH content - ↓ cellular viability Acute and chronic exposure: - Impaired glutamate metabolism in astrocytes: ↓ reuptake and metabolism, inhibition of glutamine synthetase - ↑ uptake of Ca^2+^ - ↑ accumulation of C-MeAIB	[37]
Wistar rats	Roundup^®^: 1% in drinking water (0.38% GLY)	Subchronic exposition: from GD5 to PND21. Chronic exposition: from GD5 to PND60.	Investigate the effects of subchronic exposure to GBH on neurochemical and behavioral parameters in immature and adult offspring	- ↓ uptake and ↑ glutamate release - Activation of NMDARs and influx of Ca^2+^ - GLY can bind to the glutamate and glycine cavities of NMDAR - ↓ AChE activity in offspring - Depressive behavior - ↓ levels of GSH (acute exposure) - ↓ GST after acute exposure, ↑ after chronic exposure - ↓ SOD and G6PD activity - ERK1/2 overactivation - ↓ NF-kB activation - ↓ levels of S100B protein during development, which later increased in adult offspring	[3]
Wistar rats	Roundup^®^: 70 mg/kg/day orally	Subchronic exposition: from GD5 to PND15.	Investigate possible biochemical and cell-persistent effects in the brain of adult rats following perinatal exposure to GBH	- Changes in the peptide expression in the SN - ↓ expression of peptides from the dynorphin family - ↓ dynorphin immunoreactivity in the SN and hippocampus - ↑ number of nestin-positive hippocampus cells	[66]
Wistar rats	GLY: 24 or 35 mg/kg intraperitoneally	Dams received injections every 48 h from GD8 to GD20, totaling seven injections over two weeks	Evaluate the neurobehavioral effects of GLY in neonate rats after gestational exposure	- Delay in the development of neonatal reflexes in offspring - ↓ locomotion - Learning and memory deficits - Negative regulation of the Wnt5a/CaMKII pathway	[67]
Sprague-Dawley rats	GLY, Roundup^®^: 5 mg/kg/day orally	From GD10 to PND22	Compare the potential effects of a low dose of GLY and GBH on maternal behavior and maternal neuroplasticity, focusing on the hippocampus and cingulate gyrus	- Initial reduction in maternal licking behavior, followed by a subsequent increase - Impairment of neurogenesis and plasticity in the mother’s hippocampus - Alteration in synaptophysin expression	[68]
CF-1 mice	Glifloglex^®^: 50 mg/kg/day intranasally	Four weeks (three injections per week)	Elucidate the mechanisms by which the intranasal administration of a GBH exerts its neuropathological effects	- ↓ total thiol content and CAT activity - ↓ expression of ChAT and α7 nAChRs and AChE activity - ↑ number of astrocytes - ↓ GPT and GOT transaminase activity	[69]
Sprague-Dawley rats	GLY: 50, 100, or 150 mg/kg intraperitoneally	Two weeks (three injections per week)	Assess the integrity of the nigrostriatal and mesolimbic dopaminergic systems and their relationship with spontaneous locomotor activity after repeated or acute exposure to GLY	- ↓ locomotion in the short term - ↓ specific binding of an antagonist to dopamine D1 receptors in the short term - ↓ levels of extracellular dopamine in the short term	[70]
ICR mice	Roundup^®^: 50 mg/kg/day orally	From GD14 to PND7	Assess the miRNA expression patterns in the PFC of mouse offspring after exposure to GBH during pregnancy and lactation	- Dysregulation of 53 miRNAs involved in brain development and in the pathogenesis of non-destructive diseases	[71]
Balb/c mice	Roundup^®^: 25, 50 or 100 mg/kg orally	Acute exposure	Investigate the behavioral effects induced by acute exposure to a GBH in increasing doses	- ↓ exploratory capacity of females - ↑ immobility time	[72]
Wistar rats	GBH: 2.5, 5, 10, 20 or 40 mM	Single dose	Assess the inhibitory potency of a GBH on AChE activity in rat tissues	- GBH is a weak inhibitor of AChE activity	[73]
Wistar rats	GLY: 35 or 70 mg/kg subcutaneous injection	From PND7 to PND27	Evaluate the effects of glyphosate on hippocampal synapses and cognitive functioning	- Impairment of spatial memory and recognition - ↓ expression of CAMKII - ↓ expression of synaptic proteins	[74]
Wistar rats	GLY: 35, 75, 150 or 800 mg/kg/day orally	Six days	Determine the effects of GLY on the levels of DA, NE, and 5-HT and their metabolites, as well as the turnover in striatum, hippocampus, PFC, hypothalamus, and midbrain.	- Significant dose- and region-dependent decreases in 5-HT, DA, and NE contents - ↑ turnover of 5-HIAA/5-HT in the striatum and DOPAC+HVA/DA in the PFC and hippocampus - ↓ turnover of NE/MHPG in the PFC and hypothalamus	[13]

Abbreviations: GBH, glyphosate-based herbicide; ↓, decrease; ↑, increase; 5-HT, serotonin; TH, tyrosine hydroxylase; AChE, acetylcholinesterase; SOD, superoxide dismutase; GD, gestational day; PND, postnatal day; TNF-α, tumor necrosis factor alpha; BDNF, brain-derived neurotrophic factor; TrkB, tyrosine-related kinase receptor B; NMDAR: N-methyl-D-aspartate receptor; L-VDCC, voltage-dependent calcium channels; CaMKII, Ca2+/calmodulin-dependent protein kinase II; ERK, extracellular signal-regulated kinases; LPO, lipid peroxidation; GGT, gamma-glutamyl transferase; G6PD, glucose-6-phosphate dehydrogenase; GSH, glutathione; C-MeAIB, C-methylaminoisobutyric acid; GLY, glyphosate; GST, glutathione S-transferase; NF-kB, nuclear factor-kB; S100B, S100 calcium-binding protein B; SN, substantia nigra; CAT, catalase; ChAT, choline acetyltransferase; nAChRs, nicotinic acetylcholine receptors; GPT, glutamate-pyruvate transaminase; GOT, glutamate-oxaloacetate transaminase; PFC, prefrontal cortex; DA, dopamine; NE, noradrenaline; 5-HIAA, 5-hydroxyindoleacetic acid; DOPAC, 3,4-dihydroxyphenylacetic acid; HVA, homovanillic acid; MHPG, methoxy-4-hydroxyphenylglycol.

**Table 3 ijms-23-04605-t003:** In vitro effects of glyphosate and/or its commercial formulations in rodents.

Cellular Line	Dose and Time of Exposure	Objectives	Results	Reference
PC12 cells	GLY: 0, 5, 10, 20, or 40 mM for 12, 24, 48, or 72 h	Investigate the neurotoxicity of GLY in differentiated rat PC12 cells and explore the role of apoptosis and autophagy pathways in toxicity	- ↓ cell viability - Activation of autophagic and apoptotic cell death pathways	[75]
Hippocampal pyramidal cells	GLY: 0.5 or 1 mg/mL for five or ten days	Examine the effects of glyphosate on synapse formation and maturation in the hippocampus	- ↓ dendritic complexity and synaptic column formation and maturation - ↓ synapse formation in hippocampal neurons	[74]
NSC	GLY: 0.1, 700, 7000, or 36,000 μg/L for 24 h	Understand the effects of two maximum permissible concentrations of GLY on the basic processes of neurogenesis in NSCs of the postnatal mouse subventricular zone.	- ↓ cell viability and induction of cytotoxicity - ↓ cell migration and differentiation - ↓ expression of neuronal and astrocytic genes - ↓ expression of the *CYP1A1* gene - ↑ expression of the *SOD1* gene - ↑ Ca^2+^ signaling	[76]
Astroglioma (C6)	GLY: concentrations from 0 to 160 mM for 24 h	Determine the activity of enzymes related to energy metabolism, as well as parameters of oxidative stress, mitochondrial mass, nuclear area, and autophagy in astrocytes treated with GBH	- ↓ cell viability - ↓ in the activity of the enzymes of the mitochondrial respiratory chain - ↓ CK activity - ↓ mitochondrial mass - ↑ non-protein thiol levels - ↑ autophagic protein levels	[77]
Embryonic DRG and pure Schwann cells	GLY, Roundup^®^: 0.0005% and 0.005% for ten days (DRG) or 72 h (Schwann cells)	Investigate the effects of pure GLY and GBH in murine embryonic DRG cultures	- GBH had a demyelinating effect, but this effect was not observed after treatment with GLY - GBH ↑ expression of TNF-α in DRG and in Schwann cells - GBH ↑ NO release in Schwann cells	[78]
Embryonic DRG and pure Schwann cells	GLY, Roundup^®^: doses not specified for ten days (DRG) or 72 h (Schwann cells)	Study and compare the effects of pure GLY and GBH in murine embryonic DRG explant cultures	- GBH had a concentration-dependent demyelinating effect - GBH ↑ TNF-α expression and NO release in Schwann cells	[79]

Abbreviations: PC-12, pheochromocytoma; GLY, glyphosate; ↓, decrease; NSC, neural stem cells; ↑, increase; SOD, superoxide dismutase; GBH, glyphosate-based herbicide; CK, creatine kinase; DRG, dorsal root ganglia; TNF-α, tumor necrosis factor alpha; NO, nitric oxide.

**Table 4 ijms-23-04605-t004:** Effects of glyphosate and/or its commercial formulations in fish.

Species	Dose and Time Exposure	Objectives	Results	Reference
*Cnesterodon decemmaculatus* *(Ten-spotted livebearer*)	GLY: 1 or 10 mg/L for 96 h	Assess the effect of seasonal variability on AChE activity in fish exposed to chlorpyrifos and GLY	- ↓ AChE activity dose-dependently (all seasons) - Fish were more susceptible to GLY in autumn - No inhibitory effect on AChE was observed when in vitro tests were performed at a wide range of GLY concentrations	[132]
*Colossoma macropomum* (Blackfin pacu)	Roundup^®^: 10 or 15 mg/L for 96 h	Investigate the effects of GBH on gill morphology and function, hematological parameters, biotransformation enzymes, the antioxidant system in the gills and liver, as well as on both neurological and erythrocytic DNA damage	- ↓ AChE activity	[133]
*Danio rerio* (Zebrafish)	Roundup^®^, GLY: 0.01, 0.065, or 0.5 mg/L for 96 h	Evaluate the effects of GLY and GBH on morphological and behavioral parameters in larvae and adult zebrafish	GLY and GBH caused: - Alteration of locomotion and aversive behavior in larvae - ↓ locomotion and aggressive behavior in adults - GBH caused memory impairment in adults	[134]
*Danio rerio* (Zebrafish)	GLY: 5, 10, or 50 μg/mL for 96 h	Identify a possible mechanism of toxicity for GLY related to changes in microtubule stability, which could alter the distribution and dynamics of cytoskeletal components	- GLY (50 μg/mL) ↓ levels of acetylated α-tubulin - GLY (10 and 50 μg/mL) ↓ percentage of polymeric tubulin There was no impairment of the stability of the actin filaments or the expression patterns of α-tubulin	[135]
Pintado da Amazônia	Roundup^®^: 0.37, 0.75, 2.25, 4.5, 7.5, 11.25, 15, 22.5, or 30 mg/L for 24, 48, 72, or 96 h	Evaluate the lethal concentration of the GLY and the oxidative stress parameters in tests with sublethal concentrations	- ↑ content of carbonyl protein - Brain LPO levels remained unchanged - ↑ brain AChE activity	[136]
*Danio rerio* (Zebrafish)	GLY: 0.3 or 3 μg/L for 2 weeks	Analyze the neurotoxicity of GLY in adult zebrafish after exposure through water to environmentally relevant concentrations	- Deterioration of exploratory and social behaviors, with increased anxiety - ↑ levels of 5-HT in the anterior brain - ↑ DA, DOPAC, and HVA in the anterior brain; ↑ DOPAC/DA and HVA/DA turnover - Downregulation of the expression of genes involved in the dopaminergic system: *th1*, *th2*, *comtb*, and *scl6a3* - ↑ LPO and alteration of brain antioxidant status: ↑CAT and SOD activity and ↓ GSH	[130]
*Danio rerio* (Zebrafish)	GLY: 0.05 to 10.000 μg/L for a period of 1.5 to 120 h after fertilization	Explore the effects of the use of different concentrations of GLY on anatomy and behavior of fish	High concentrations of GLY (≥1000 μg/L) caused: - Electrophysiological changes in the midbrain and ↓ locomotion High and low concentrations of GLY were associated with: - Morphological signs of microglia activation - Dysregulation of genetic pathways involved in neuronal physiology, synaptic transmission, and inflammation - Absence of neurovascular structural malformations	[137]
*Hypomesus transpacificus* (Delta smelt)	Roundup^®^: 0.064, 0.64, 6.4, 64, or 640 mg/L for 6 h	Compare the sublethal toxicity of four herbicides (penoxsulam, imazamox, fluridone, and GBH).	- GBH did not cause alterations in AChE activity	[138]
*Cyprinus carpio* (European carp)	GLY: 0.02, 0.05, 0.07, or 0.1 mg/L for 24, 48, 72, or 96 h	Analyze the effect of GLY and atrazine on the hematological and biochemical parameters of blood and on behavioral aspects	- GLY caused behavioral disturbances	[139]
*Danio rerio* (Zebrafish)	Roundup^®^: 2, 5, or 8.5 μg/mL for 72 h	Investigate the lethal and sublethal developmental effects, neurotoxic potential, and oxidative stress responses after GBH exposure	High concentrations of GBH caused: - Developmental toxicity - Malformations Low concentrations of GBH: - Did not have teratogenic effects and did not alter oxidative stress, neurotransmission, or the regulation of energy metabolism - Did not cause histopathological changes in the brain	[140]
*Danio rerio* (Zebrafish)	Roundup^®^: 1, 2, or 5 μg/mL for 72 h	Assess GBH effects at environmentally relevant concentrations through a set of behavioral patterns	- Did not alter exploratory or social behavior but did induce changes in avoidance behavior Cortisol levels were not altered	[141]
*Carassius auratus* (Goldenfish)	Nongteshi^®^: 0.22, 0.44, or 0.88 mmol/L for 96 h	Investigate the toxic effects of GBH exposure using a metabolomic approach supplemented with histological inspection and hematological evaluation	- Destruction of the microscopic structure of the brain - High concentrations of GBH caused behavioral disturbances - Alteration of the balance of neurotransmitters: ↓ glutamate and GABA levels, ↑ glutamine levels - ↓ levels of the glial marker myoinositol and NAA - ↓ levels of brain creatine/phosphocreatine	[142]
*Carassius auratus*	Nongteshi^®^: 0.2 mmol/L for 90 days	Assess GBH toxicity after prolonged exposure	Results in the brain: - ↓ choline, phosphocholine, and betaine levels - ↑ 3-hydroxybutyrate levels - ↓ AMP levels - ↓ succinate and citrate - Positive correlations between choline and phosphocholine, choline and betaine, glycine, and sarcosine	[143]
*Danio rerio* (Zebrafish)	GLY: 5 or 10 mg/L for 24 and 96 h	Evaluate oxidative stress parameters, as well as the activity and expression of AChE	- ↓ brain LPO 24 h after exposure to GLY (10 mg/L) - No alterations in the generation of ROS - No alterations in AChE activity were detected AChE gene expression in the brain decreased after 24 h for both GLY concentrations and improved 96 h after exposure to 10 mg/L GLY	[144]
*Cnesterodon decemmaculatus*	GLY: 1, 17.5, or 35 mg/L for 96 h	Assess the toxic effect of acute exposure to sublethal GLY concentrations on AChE activity in different parts of the body	- ↓ AChE activity in the anterior and middle sections of the body but not in the posterior section	[145]
*Odontesthes bonariensis* (Argentinian silverside)	GBH: 1 or 10 mg/L for 15 days	Determine the basal levels of adenylates, phosphagens, and the AEC index in the brain, muscle, and liver, as well as the impact of exposure to sublethal GBH on the subcellular energy balance	- There was no alteration of cerebral AEC	[146]
*Danio rerio* (Zebrafish)	GBH, GLY: 0.065, 1, 10, 160, 1.6 × 10^3^, 4 × 10^3^, or 8 × 10^3^ mg/L for 3 h (in vitro) or for 7 days (in vivo)	Investigate the neurotoxic effects of GBH by focusing on acute toxicity, activity, and transcription levels of mitochondrial respiratory chain complexes, mitochondrial membrane potential, reactive species formation, and behavioral repertoire	In vivo exposure to GBH (7 days) caused: - Modulation of gene expression related to mitochondrial complexes - Increased ROS production - Mitochondrial hyperpolarization in brain cells - Behavioral disturbances Low concentrations of GBH (0.065, 1 mg/L) caused: - Inhibition of NADH dehydrogenase and cytochrome C	[147]
*Danio rerio* (Zebrafish)	Roundup^®^, GLY: 50 μg/mL for 24 h	Investigate the neurotoxic effects of GBH and GLY exposure on the developing brain	Both GBH and GLY caused: - Loss of delineated cerebral ventricles and reductions in the cephalic and ocular regions ↓ gene expression in the forebrain, midbrain, and eye, but no changes were detected in the rhombencephalon	[148]
*Jenynsia multidentate* (Onesided livebearer)	Roundup^®^ (Original, Transorb or WG): 0.5 mg/L for 96 h	Evaluate and compare the effects of three GBH formulations on behavior patterns	Roundup WG^®^ was the most harmful formulation and negatively affected: - Social interaction - Space exploration and swimming performance - Long-term memory consolidation Roundup Transorb^®^ had more severe effects on sexual behavior	[149]
*Rhamdia quelen* (Silver catfish)	GLY: 6.5 mg/L for 12, 24, 48, or 72 h	Investigate the effects of GLY on the antioxidant system, as well as the neurotoxic effects on eggs and larvae	- ↑ AChE activity at 12 and 48 h of treatment, ↓ between 48 and 72 h - There were no changes in CAT or GST activity - ↑ activity of the GR at 12 and 24 h - ↑ LPO at 48 h of treatment	[150]
*Danio rerio* (Zebrafish)	GLY: 0.01, 0.1, 0.5, 1, 5, 10, 100, 200, 400, or 600 mg/L from 3 hpf until 96 hpf	Assess the developmental, morphological, and genetic effects of GLY in zebrafish embryos	- Concentrations of GLY higher than 100 mg/L caused delay and alterations in development, as well as embryonic death - Damage to axons of primary caudal motor neurons in embryos and increased locomotion in larvae	[151]

Abbreviations: GLY, glyphosate; ↓, decrease; AChE: acetylcholinesterase; GBH, glyphosate-based herbicide; ↑, increase; LPO, lipid peroxidation; 5-HT, serotonin; DA, dopamine; DOPAC, 3,4-dihydroxyphenylacetic acid; HVA, homovanillic acid; CAT, catalase; SOD, superoxide dismutase; GSH, glutathione; GABA, gamma-aminobutyric acid; NAA, N-acetyl-L-aspartate; AMP, adenosine monophosphate; ROS, reactive oxygen species; AEC, adenylate energy charge; NADH, nicotinamide adenine dinucleotide dehydrogenase; GST, glutathione S-transferase; GR, glutathione reductase; hpf, hours post fertilization.

## Data Availability

Not applicable.

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
