# Peer review of "Toxic Effects of Glyphosate on the Nervous System: A Systematic Review"

_ijms, 2022, doi:10.3390/ijms23094605_

Round 1

Reviewer 1 Report

In this manuscript, the authors review the recent literature regarding the neurotoxicity of glyphosate and glyphosate-based herbicides. Overall, I found this article to be very interesting. It is well written and well organized. I have a few general comments that I feel would help.
The authors need to be more careful in their use of the terms pesticide and herbicide. For this manuscript, I recommend that they only use those terms when talking about general pesticides or herbicides. If they are referring to glyphosate-based herbicides, they need to be specific. Sometimes it is confusing to know whether they are referring to a pesticide in general, or if it is specifically glyphosate. I believe this needs to be addressed throughout the manuscript.
At times, the authors use terms that a very vague such as “significant”, “a long time” and “well below.” I would recommend being more specific, if possible. Specifically, I feel strongly that the authors need to be more specific about what is considered a long time for glyphosate to persist in the environment. The statements regarding the environmental persistence need to have additional references, as this is a key point to any environmental toxicant. From my understanding, some reports show the environmental half life of glyphosate to be as short as one day. This would not be considered a long time. However, other reports suggest that the environmental half life of glyphosate to be more than 150 days, which is a long time, and is nearing that of some chlorinated hydrocarbon pesticides. However, some reports indicate that when the half life is that long, it is because it is bound in such as way that it cannot be metabolized and is thus non toxic. This is a very important point, one that I feel deserves a little more attention in the introduction.
The authors allude to the fact that there may be bioaccumulation of glyphosate in fish. This could be a very important aspect of glyphosate toxicology, and should be expanded if more is known.
The first sentence of the introduction needs to be referenced. This is a big statement and should have references to support it.
From what I understand, Bayer is going to discontinue production of glyphosate. Are there any predictions as to how this will impact the global market of glyphosate-based herbicides?
Figure 1 along with the text explaining the criteria for either exclusion or inclusion of the literature is somewhat confusing, vague, or inconsistent. For example, the text indicates that studies that used glyphosate doses above the NOAEL were excluded. However, in the table you indicate that 4 articles were excluded because the glyphosate dose was below the NOAEL. Also, I am curious as to why articles were included that were in English and Spanish. How many articles were in Spanish? What is the quality of those journals? Should they be included as much of the readership of the journal will not be able to read those references. The math is inconsistent, in the text you indicate that 163 articles were screened with 88 of them being excluded. This would leave 75 articles, but you indicate that there are 51 articles that were included in the review. Please revise the text. It appears to be correct in the figure. Additionally, in the figure you indicate that 48 of the 88 articles were not relevant. Why were they not relevant? Why were case studies not included? Finally, I understand why articles that reported the effects of multiple pesticides were not included, however, that mixture toxicology could be even more important that the effects of glyphosate alone (additive or synergistic effects).
In section 3.1.1, you state that the glyphosate may be less to toxic to farmers than other pesticides, and that future research needs to evaluate this. I would assume that after 50 years of use, it should be well known if glyphosate is less toxic to farmers than other similar herbicides.
In the last paragraph of section 3.1.2, you indicate that LDH is produced in almost every organ in the body, which is correct. However, in the next sentence you indicate that increases in serum LDH are indicate of CNS effects. The logic here does not make sense. Please modify and clarify.
I really like the fact that you note that one of the limitations of the study is the potential effects of the non-glyphosate components of glyphosate-based herbicides in causing toxicity. There is a recent paper published in the March 2022 issue of Toxicological Sciences demonstrating that Roundup has greater genotoxicity than glyphosate alone. Perhaps this article should be included to reference the potential confounding effects of the non-glyphosate components of glyphosate-based herbicides.

Author Response

  1. The authors need to be more careful in their use of the terms pesticide and herbicide. For this manuscript, I recommend that they only use those terms when talking about general pesticides or herbicides. If they are referring to glyphosate-based herbicides, they need to be specific. Sometimes it is confusing to know whether they are referring to a pesticide in general, or if it is specifically glyphosate. I believe this needs to be addressed throughout the manuscript.

Indeed, the use of the terms herbicide and pesticide to refer both to these substances in general and to glyphosate can be doubtful and lead to mistakes. Therefore, in accordance with the reviewer's recommendation, we have modified in the text all the terms “herbicide” and “pesticide” referring to glyphosate and its commercial formulations for more precise and adequate terms.

  1. At times, the authors use terms that a very vague such as “significant”, “a long time” and “well below.” I would recommend being more specific, if possible.

In accordance with the reviewer's comment, we have replaced the general terms “significant”, “a long time” and “well below” throughout the manuscript with other more specific terms.

  1. Specifically, I feel strongly that the authors need to be more specific about what is considered a long time for glyphosate to persist in the environment. The statements regarding the environmental persistence need to have additional references, as this is a key point to any environmental toxicant. From my understanding, some reports show the environmental half life of glyphosate to be as short as one day. This would not be considered a long time. However, other reports suggest that the environmental half life of glyphosate to be more than 150 days, which is a long time, and is nearing that of some chlorinated hydrocarbon pesticides. However, some reports indicate that when the half life is that long, it is because it is bound in such as way that it cannot be metabolized and is thus non toxic. This is a very important point, one that I feel deserves a little more attention in the introduction.

According to reviewer's comment, we also feel it is important to expand and clarify the information regarding how long glyphosate persists in the environment. As the reviewer points out, the available literature on the persistence of glyphosate in the environment is very diverse and supports very different periods of permanence. We have carried out an exhaustive search on this subject and the conclusions that we have drawn and have tried to synthesize in our review is that, in general, the persistence of glyphosate in the different environmental compartments (such as soil or water) is low or moderate. This seems to be mainly due to the degradation of glyphosate carried out by the microorganisms that inhabit these regions and/or its union with the particles that make up the soil. However, some studies indicate that these processes do not fully and definitively “inactivate” glyphosate.

On the one hand, the main product of glyphosate degradation is aminomethylphosphonic acid (AMPA), which has similar properties to glyphosate in terms of environmental behavior and toxicity but can be much more persistent. Therefore, the microbial degradation of glyphosate does not always spell the end of its toxicity. In addition, this degradation capacity of microorganisms seems to be progressively reduced with repeated applications of glyphosate, which could end up favoring its accumulation in the environment and in food products.

On the other hand, in the past glyphosate was not considered a problem for groundwater and surface water as it tends to adhere to soil components, so its potential for mobility through soil is relatively low. However, this adsorption may not be permanent, since the available evidence suggests the possibility of a slow remobilization of these residues because of rain, so that parts of the glyphosate and its metabolite AMPA would end up contaminating groundwater. In addition, rain and erosion can also carry glyphosate- and AMPA-containing soil particles into surface water. Therefore, the available information seems to indicate that even when glyphosate or AMPA remain biologically inactive in the environment for long periods of time, this "inactivation" may not be definitive and may end up affecting the health of nearby organisms.

In summary, although most glyphosate is rapidly degraded or inactivated, this compound can remain in small amounts in soil, water, and food. As a result of their intensive use on a large scale, glyphosate and AMPA could accumulate in the environment and in food products, leading to a variety of detrimental side effects for soil and water quality, as well as animal and human health.

In response to this comment, we have expanded the information regarding the persistence of glyphosate in the environment. In this way, we have tried to reflect the most important aspects previously discussed with the following paragraph, please consider it:

In general, in the different environmental compartments, glyphosate is mainly degraded by microorganisms, so that its persistence is considered to be low to moderate, although it is considerably variable. On the one hand, although glyphosate is assumed to be readily degraded in soil, its biodegradation is influenced by numerous factors, including physico-chemical, biological properties and soil composition (Kanissery et al., 2019). Thus, the half-life of glyphosate in soil can range from 1 to 280 days, while that of AMPA, its main metabolite, ranges from 23 to 958 days (Bento et al. 2016; Bergström et al. 2011; Laitinen et al. 2006). In soil, glyphosate can bind strongly to its constituent particles and remain biologically inactive, or it can reach groundwater, due to its high water solubility (Tzanetou and Karasali 2020). However, repeated applications of glyphosate have been shown to result in a gradual difficulty for its biodegradation in the soil, which could increase the risk of groundwater contamination (Andréa et al. 2003; Carretta et al. 2021). In water, the permanence of glyphosate is also widely variable and depends on factors such as light and temperature, being more persistent and toxic under conditions of darkness and higher water temperatures (Leoci and Ruberti 2020). In general, the half-life of glyphosate in water varies from a few days to 91 days (Battaglin et al. 2014; Castro Berman et al. 2018), although it has been found to remain for up to 315 days in marine waters (Mercurio et al. 2014). On the other hand, while the persistence of glyphosate in vegetation may be only days, several studies have detected its presence in many foods and crops even a year after application (EFSA 2018; Edge et al. 2021).

Therefore, although the concentrations of glyphosate residues that persist over time are relatively low, it is possible that due to extensive use on a large scale they may accumulate and become a risk to animal and human health, as they are chronically exposed to residues in the water and food they consume (Battaglin et al. 2014; Niemann et al. 2015; Reddy et al. 2008). This has been confirmed by the detection of glyphosate in the organs and urine of a high proportion of farm animals and farmers (Acquavella et al. 2004; Conrad et al. 2017; Krüger et al. 2014a, b). In addition, residues were also found in the urine of 60-80% of the general population in the United States at medium and maximum concentrations of 2-3 and 233 μg/L, respectively. In Europe, residues were also detected in the urine of 44% of the population, although their average and maximum concentrations were lower: <1 and 5 μg/L, respectively (Niemann et al. 2015; Van Bruggen et al. 2018).”

  1. The authors allude to the fact that there may be bioaccumulation of glyphosate in fish. This could be a very important aspect of glyphosate toxicology and should be expanded if more is known.

In response to the reviewer's comment, we have searched for more recent articles related to the bioaccumulation capacity of glyphosate in fish, since the only article that we had included in the review dates from 1994 (Wang et al. 1994). Although some of the articles identified (mainly reviews) defend that, given the physicochemical characteristics of glyphosate, it is expected that it cannot accumulate in fish, we have also identified new evidence that supports the bioaccumulation capacity of this compound (alone or in commercial formulation) in some aquatic species (Contardo-Jara et al. 2009; Dey et al. 2016; El-Sheshtawy et al. 2021; Rossi et al. 2020). Although it is true that the capacity for bioaccumulation has been demonstrated in only some species of fish, we believe it is important to refer to the possibility that this accumulation may also occur in other animal and marine plant species. This could lead to biomagnification of glyphosate throughout the aquatic food chain which could ultimately pose a health risk to humans. Thus, in the manuscript, we have modified the sentence on the bioaccumulation capacity in fish, so that we have tried to include three important aspects: 1) considering the physical-chemical characteristics of glyphosate, it would not be expected to accumulate in fish; 2) this bioaccumulative capacity has been found in some aquatic species, but not in all; and 3) that there is recent evidence supporting this ability to bioaccumulate. To reflect this information we have included the following sentence in the manuscript, please consider it:

In addition, although given the physicochemical characteristics of glyphosate it is not expected that it can bioaccumulate in the tissues of animals, the available evidence supports that this compound can accumulate in the tissues of some aquatic species, which could mean an increase in its neurotoxicity (Contardo-Jara et al. 2009; Dey et al. 2016; El-Sheshtawy et al. 2021; Rossi et al. 2020)”.

  1. The first sentence of the introduction needs to be referenced. This is a big statement and should have references to support it.

Based on the reviewer's comment, we have added the reference from where we have obtained that affirmation.

  1. From what I understand, Bayer is going to discontinue production of glyphosate. Are there any predictions as to how this will impact the global market of glyphosate-based herbicides?

Indeed, Bayer confirmed that it will stop selling glyphosate-based herbicides for residential (lawn and garden) use in the EUA. To manage the risk of litigation, the company said it will reformulate Roundup® herbicide in 2023, replacing the glyphosate for other active ingredients that have not yet been disclosed. However, the measure is aimed exclusively at residential use, as glyphosate-based herbicides will continue to be available for professional and agricultural uses.

This measure may be beneficial in terms of the health of individuals who regularly use glyphosate-based herbicides (GBH) in their homes, since it will be carried out precisely to face the numerous lawsuits against the company that maintain that the continued use of these products has caused cancer in users. The problem is that as large-scale agricultural use of glyphosate continues, farmworkers remain at risk of experiencing its harmful effects. For example, in two of the human studies included in our review, visual memory impairment was detected in farmers and the highest incidence of developmental disorders was documented in children whose mothers lived in regions close to farming areas. Furthermore, based on the many cases in which residential use of GBH has been linked to serious health problems in users, it is likely that farmers are especially vulnerable to experiencing more and more serious health problems, as they are exposed to larger quantities of the product and for a longer period.

On the other hand, according to our knowledge, the commercialization and extensive use of GBH in farming areas will continue for an indefinite period of time and with it its arrival in nearby aquatic ecosystems. This situation maintains the risk of exposure and intoxication of the organisms present both in the soil and in the aquatic environments of these regions. Therefore, while ceasing the marketing of Roundup® for home use may be a first step in reducing GBH exposure and toxic effects, its continued use in professional and agricultural settings will continue to pose a health risk to animals and humans.

  1. Figure 1 along with the text explaining the criteria for either exclusion or inclusion of the literature is somewhat confusing, vague, or inconsistent. For example, the text indicates that studies that used glyphosate doses above the NOAEL were excluded. However, in the table you indicate that 4 articles were excluded because the glyphosate dose was below the NOAEL.

Indeed, we have detected the presence of some mistakes in the PRISMA diagram in Figure 2 that we have already corrected. In this way, in the diagram, we have modified the position of the articles that have been eliminated because they have not been published in the last ten years, the case studies, and the studies that used doses above the NOAEL. In the original document, these reasons for exclusion appeared in the "eligibility" section of the diagram, however, we have relocated them to the "screening" section, since the information necessary to exclude them was obtained by reading only the title and abstract. On the other hand, as the reviewer points out, there was a contradiction between the text and Figure 2 regarding the exclusion criterion referring to the NOAEL. In line with this, we have corrected the error present in Figure 2 to indicate that the articles that have been excluded were those that used a dose of glyphosate greater than the NOAEL.

  1. Also, I am curious as to why articles were included that were in English and Spanish. How many articles were in Spanish? What is the quality of those journals? Should they be included as much of the readership of the journal will not be able to read those references.

The main objective for including articles in Spanish was to try to identify the greatest possible number of articles and, in this way, be able to draw more rigorous conclusions by having a greater volume of evidence. However, in this work no article in Spanish has been included, since all the works found and that meet the inclusion criteria are in English. However, we value this point made by the reviewer very positively, so that possibly in our following reviews we will only include the articles in English.

  1. The math is inconsistent, in the text you indicate that 163 articles were screened with 88 of them being excluded. This would leave 75 articles, but you indicate that there are 51 articles that were included in the review. Please revise the text. It appears to be correct in the figure.

Based on the reviewer's comment, we have corrected the text of the methodology section as follows, please consider:

In accordance with the inclusion and exclusion criteria, 163 titles and abstracts were screened to verify if they met the criteria previously mentioned. After this procedure, 112 articles were excluded for the reasons summarized in Figure 1. Finally, 51 articles were included in the present systematic review”.

  1. Additionally, in the figure you indicate that 48 of the 88 articles were not relevant. Why were they not relevant?

In the reason for exclusion "articles not relevant to our study" we have included all those articles that did not focus on identifying the possible effects and mechanisms of action of glyphosate on the nervous system, but that could not be included in the exclusion criteria previously established.

For example, in the databases there are some articles available referring to the protective capacity of certain substances against the neurotoxic alterations induced by glyphosate. Although these articles are very interesting, having included them in our review would have meant having to handle a very large number of studies and, consequently, the analysis of each of them would have had to be much less detailed and in-depth.

  1. Why were case studies not included?

Although in our first version of the manuscript we have included the case studies, we finally decided to exclude them from this review. The main reason for this exclusion (and not for other types of human studies) was because virtually all of the identified case studies described suicide attempts. In these articles, the effects of oral exposure to very high doses of glyphosate or GBH are described. Therefore, the amounts of glyphosate these people had been exposed to are not representative of the doses to which the general population and/or farmers may be exposed. However, the most important findings obtained in these studies are included in our discussion, in which we comment on the ability of glyphosate to cross the blood-brain barrier and its predisposition to cause important dysfunctions in memory, alterations that are sometimes irreversible.

  1. Finally, I understand why articles that reported the effects of multiple pesticides were not included, however, that mixture toxicology could be even more important that the effects of glyphosate alone (additive or synergistic effects).

Indeed, we are aware that in “real life” animals and humans are more likely to be exposed to mixtures of different classes of pesticides than to one type of pesticide. Although the study of the toxic effects of the mixture of pesticides is extremely useful, we consider that knowing in depth the effects and mechanisms of action of glyphosate individually is a necessary previous step to achieve a better interpretation of the effects of a possible exposure to a mixture of pesticides. Specifically, we believe that knowing the specific effects produced by each pesticide individually will help distinguish, in cases of poisoning by multiple pesticides, which effects have already been attributed to the action of a specific pesticide and which are novel and, therefore, they could be a consequence of the interaction between various pesticides. However, the neurotoxic effects derived from exposure to multiple pesticides constitute a very interesting and useful topic, and one that is already being addressed by some research groups.

  1. In section 3.1.1, you state that the glyphosate may be less to toxic to farmers than other pesticides, and that future research needs to evaluate this. I would assume that after 50 years of use, it should be well known if glyphosate is less toxic to farmers than other similar herbicides.

Although it would be reasonable to have sufficient information on the potential toxic effects of glyphosate on the health of farmers, the reality is that in selecting the articles for this review we appreciate that the available evidence on this subject is very limited.

The statement that glyphosate might be less toxic to farmers than other pesticides is based solely on two studies by the same authors: Zang et al. (2016, 2018). In both studies, the authors primarily evaluated the effect of glyphosate on peripheral nerve conduction in a sample of Chinese farmers. However, these studies had several limitations that must be considered when generalizing the results: they used relatively small samples, limited to a single geographical region (China), and the follow-up was carried out for a short period of time. It is possible that the sampled farmers used too low amounts of glyphosate that are not representative of those used in other parts of the world. The short duration of the investigations could also limit the interpretation of the potential abnormalities on peripheral nerve conduction since this neurological effect can be chronic. Furthermore, although both investigations focused on the study of the effects of glyphosate on peripheral nerve conduction, there are many other aspects of the farmers' health that were not taken into consideration. For example, there is numerous evidence in animals and humans that points to the hippocampus (and thus the memory) as one of the main targets of glyphosate's action. However, no evaluation was conducted in these studies that looked at the effect of glyphosate on the cognitive functions of farmers. Therefore, it is highly possible that the two studies by Zhang et al. (2016, 2018) produced biased results and, therefore, we cannot consider them axiomatic in drawing conclusions about the general effects that glyphosate exerts on farmers.

The problem is that in our review we have only been able to identify four studies that evaluated the effects of occupational exposure to glyphosate, which shows that the information available is insufficient. Therefore, the dearth of evidence in this area led us to assume that glyphosate might be less toxic than other pesticides, based on the claim made by the Zhang et al. in both articles mentioned above.

Therefore, the limited evidence in this area led us to assume that glyphosate might be less toxic than other pesticides, based on the statement made by the team of Zhang et al. in both articles mentioned above. However, in the other two articles analyzed, occupational exposure to glyphosate was related to memory alterations and a higher probability of incidence of developmental disorders in children. It is for this reason that we believe it is necessary to carry out more research on the effects that occupational exposure to glyphosate may have on the health of farmers, especially those who have been using this class of substances for more years. In addition, we also consider it necessary to include in these investigations people who live in regions close to the farming areas, with special attention to the effects of glyphosate on the health of pregnant women and children.

  1. In the last paragraph of section 3.1.2, you indicate that LDH is produced in almost every organ in the body, which is correct. However, in the next sentence you indicate that increases in serum LDH are indicate of CNS effects. The logic here does not make sense. Please modify and clarify.

This affirmation is due to the fact that this increase in LDH was observed by Martínez et al. (2020) in SH-SY5Y cell culture from human neuroblastoma. However, as the reviewer rightly points out, our statement can lead to the mistake of considering that the increase in LDH levels is always indicative of damage to the CNS. For this reason, we have reformulated this sentence, indicating that increased levels of LDH may be indicative of damage to different tissues, including those of the CNS. The sentence is as follows, please consider it:

Thus, the glyphosate-induced LDH increases could be indicative of the damage caused by the herbicide in the CNS”.

  1. I really like the fact that you note that one of the limitations of the study is the potential effects of the non-glyphosate components of glyphosate-based herbicides in causing toxicity. There is a recent paper published in the March 2022 issue of Toxicological Sciences demonstrating that Roundup has greater genotoxicity than glyphosate alone. Perhaps this article should be included to reference the potential confounding effects of the non-glyphosate components of glyphosate-based herbicides.

Based on the reviewer's comment, we have searched both the referenced article and additional articles in which glyphosate-based formulations were found to produce greater toxicity than glyphosate alone. To summarize the results of these studies we have included the following sentence in the limitations section, please consider it:

In addition, the results of various studies support the greater toxicity of commercial glyphosate formulations compared to glyphosate administered alone (Benachour and Séralini 2009; Mesnage et al. 2022; Pereira et al. 2018; Tsui and Chu 2003)”.

References:

Acquavella, J. F., Alexander, B. H., Mandel, J. S., Gustin, C., Baker, B., Chapman, P., and Bleeke, M. (2004). Glyphosate biomonitoring for farmers and their families: results from the Farm Family Exposure Study. Environ. Health Perspect. 112, 321-326. https://doi.org/10.1289/ehp.6667

Andréa, M. M. de, Peres, T. B., Luchini, L. C., Bazarin, S., Papini, S., Matallo, M. B., and Savoy, V. L. T. (2003). Influence of repeated applications of glyphosate on its persistence and soil bioactivity. Pesq. Agropec. Bras. 38, 1329–1335. https://doi.org/10.1590/S0100-204X2003001100012

Battaglin, W. a., Meyer, M. t., Kuivila, K. m., and Dietze, J. (2014). Glyphosate and Its Degradation Product AMPA Occur Frequently and Widely in U.S. Soils, Surface Water, Groundwater, and Precipitation. J. Am. Water Resources Assoc. 50, 275–290. https://doi.org/10.1111/jawr.12159

Benachour, N., and Séralini, G. E. (2009). Glyphosate formulations induce apoptosis and necrosis in human umbilical, embryonic, and placental cells. Chem. Res. Toxicol. 22, 97-105. https://doi.org/10.1021/tx800218n

Bento, C. P. M., Yang, X., Gort, G., Xue, S., van Dam, R., Zomer, P., Mol, H. G. J., Ritsema, C. J., and Geissen, V. (2016). Persistence of glyphosate and aminomethylphosphonic acid in loess soil under different combinations of temperature, soil moisture and light/darkness. Sci. Total Environ. 572, 301–311. https://doi.org/10.1016/j.scitotenv.2016.07.215

Bergström, L., Börjesson, E., and Stenström, J. (2011). Laboratory and lysimeter studies of glyphosate and aminomethylphosphonic acid in a sand and a clay soil. J. Environ. Qual. 40, 98–108. https://doi.org/10.2134/jeq2010.0179

Carretta, L., Cardinali, A., Onofri, A., Masin, R., and Zanin, G. (2021). Dynamics of glyphosate and aminomethylphosphonic acid in soil under conventional and conservation tillage. Int. J. Environ. Res. 15, 1037–1055. https://doi.org/10.1007/s41742-021-00369-3

Castro Berman, M., Marino, D. J. G., Quiroga, M. V., and Zagarese, H. (2018). Occurrence and levels of glyphosate and AMPA in shallow lakes from the Pampean and Patagonian regions of Argentina. Chemosphere 200, 513–522. https://doi.org/10.1016/j.chemosphere.2018.02.103

Conrad, A., Schröter-Kermani, C., Hoppe, H. W., Rüther, M., Pieper, S., and Kolossa-Gehring, M. (2017). Glyphosate in German adults–Time trend (2001 to 2015) of human exposure to a widely used herbicide. Int. J. Hyg. Environ. Health 220, 8-16. https://doi.org/10.1016/j.ijheh.2016.09.016

Contardo-Jara, V., Klingelmann, E., and Wiegand, C. (2009). Bioaccumulation of glyphosate and its formulation Roundup Ultra in Lumbriculus variegatus and its effects on biotransformation and antioxidant enzymes. Environ. Pollut. 157, 57-63. https://doi.org/10.1016/j.envpol.2008.07.027

Dey, S., Samanta, P., Pal, S., Mukherjee, A. K., Kole, D., and Ghosh, A. R. (2016). Integrative assessment of biomarker responses in teleostean fishes exposed to glyphosate-based herbicide (Excel Mera 71). Emerg. Contam. 2, 191-203. https://doi.org/10.1016/j.emcon.2016.12.002

Edge, C. B., Brown, M. I., Heartz, S., Thompson, D., Ritter, L., and Ramadoss, M. (2021). The persistence of glyphosate in vegetation one year after application. Forests 12, 601. https://doi.org/10.3390/f12050601

El-Sheshtawy, S. M., Nada, M. M., Abd Elhafeez, M. S., and Samak, D. H. (2021). Protective effect of supplementation with powdered mulberry leaves on glyphosate-induced toxicity in catfish (Clarias gariepinus). Adv. Anim. Vet. Sci9, 1718-1731. http://dx.doi.org/10.17582/journal.aavs/2021/9.10.1718.1731

European Food Safety Authority (EFSA) (2018). Review of the existing maximum residue levels for glyphosate according to Article 12 of Regulation (EC) No 396/2005. EFSA J. 16, e05263. https://doi.org/10.2903/j.efsa.2018.5263

Kanissery, R., Gairhe, B., Kadyampakeni, D., Batuman, O., and Alferez, F. (2019). Glyphosate: its environmental persistence and impact on crop health and nutrition. Plants 8, 499. https://doi.org/10.3390/plants8110499

Krüger, M., Schledorn, P., Schrödl, W., Hoppe, H. W., Lutz, W., and Shehata, A. A. (2014a). Detection of glyphosate residues in animals and humans. J. Environ. Anal. Toxicol. 4, 1-5. https://doi.org/10.4172/2161-0525.1000210

Krüger, M., Schrödl, W., Pedersen, I. B., and Shehata, A. A. (2014b). Detection of glyphosate in malformed piglets. J. Environ. Anal. Toxicol. 4, 2161-0525. https://doi.org/10.4172/2161-0525.1000230

Laitinen, P., Siimes, K., Eronen, L., Rämö, S., Welling, L., Oinonen, S., Mattsoff, L., and Ruohonen-Lehto, M. (2006). Fate of the herbicides glyphosate, glufosinate-ammonium, phenmedipham, ethofumesate and metamitron in two Finnish arable soils. Pest Manag. Sci. 62, 473–491. https://doi.org/10.1002/ps.1186

Leoci, R., and Ruberti, M. (2020). Glyphosate in agriculture: environmental persistence and effects on animals. a review. J. Agric. Environ. Int. Dev. 114, 99–122. https://doi.org/10.12895/jaeid.20201.1167

Martínez, M. A., Rodríguez, J. L., Lopez-Torres, B., Martínez, M., Martínez-Larrañaga, M. R., Maximiliano, J. E., Anadón, A., and Ares, I. (2020). Use of human neuroblastoma SH-SY5Y cells to evaluate glyphosate-induced effects on oxidative stress, neuronal development and cell death signaling pathways. Environ. Int. 135, 105414. https://doi.org/10.1016/j.envint.2019.105414

Mercurio, P., Flores, F., Mueller, J. F., Carter, S., and Negri, A. P. (2014). Glyphosate persistence in seawater. Mar. Pollut. Bull. 85, 385–390. https://doi.org/10.1016/j.marpolbul.2014.01.021

Mesnage, R., Ibragim, M., Mandrioli, D., Falcioni, L., Tibaldi, E., Belpoggi, F., ... and Antoniou, M. N. (2022). Comparative toxicogenomics of glyphosate and Roundup herbicides by mammalian stem cell-based genotoxicity assays and molecular profiling in Sprague-Dawley rats. Toxicol. Sci. 186, 83-101. https://doi.org/10.1093/toxsci/kfab143

Niemann, L., Sieke, C., Pfeil, R., and Solecki, R. (2015). A critical review of glyphosate findings in human urine samples and comparison with the exposure of operators and consumers. J. Verbrauch. Lebensm. 10, 3-12. https://doi.org/10.1007/s00003-014-0927-3

Pereira, A. G., Jaramillo, M. L., Remor, A. P., Latini, A., Davico, C. E., da Silva, M. L., Müller, Y. M. R., Ammar, D., and Nazari, E. M. (2018). Low-concentration exposure to glyphosate-based herbicide modulates the complexes of the mitochondrial respiratory chain and induces mitochondrial hyperpolarization in the Danio rerio brain. Chemosphere 209, 353-362. https://doi.org/10.1016/j.chemosphere.2018.06.075

Reddy, K. N., Rimando, A. M., Duke, S. O., and Nandula, V. K. (2008). Aminomethylphosphonic acid accumulation in plant species treated with glyphosate. J. Agric. Food Chem. 56, 2125-2130. https://doi.org/10.1021/jf072954f

Rossi, A. S., Fantón, N., Michlig, M. P., Repetti, M. R., and Cazenave, J. (2020). Fish inhabiting rice fields: Bioaccumulation, oxidative stress and neurotoxic effects after pesticides application. Ecol. Indic. 113, 106186. https://doi.org/10.1016/j.ecolind.2020.106186

Tsui, M. T., and Chu, L. M. (2003). Aquatic toxicity of glyphosate-based formulations: comparison between different organisms and the effects of environmental factors. Chemosphere 52, 1189-1197. https://doi.org/10.1016/S0045-6535(03)00306-0

Tzanetou, E., & Karasali, H. (2020). Glyphosate residues in soil and air: an integrated review. In D. Kontogiannatos, A. Kourti, and K. F.  Mendes (Eds.), Pests, weeds and diseases in agricultural crop and animal husbandry production. IntechOpen. https://doi.org/10.5772/intechopen.93066

Van Bruggen, A. H. C., He, M. M., Shin, K., Mai, V., Jeong, K. C., Finckh, M. R., and Morris, J. G. (2018). Environmental and health effects of the herbicide glyphosate. Sci. Total Environ. 616-617, 255-268. https://doi.org/10.1016/j.scitotenv.2017.10.309

Wang, Y. S., Jaw, C. G., and Chen, Y. L. (1994). Accumulation of 2,4-D and glyphosate in fish and water hyacinth. Water Air Soil Pollut. 74, 397-403. DOI: 10.1007/BF00479802

Zhang, C., Hu, R., Huang, J., Huang, X., Shi, G., Li, Y., Yin, Y., and Chen, Z. (2016). Health effect of agricultural pesticide use in China: implications for the development of GM crops. Sci. Rep. 6, 34918. https://doi.org/10.1038/srep34918

Zhang, C., Sun, Y., Hu, R., Huang, J., Huang, X., Li, Y., Yin, Y., and Chen, Z. (2018). A comparison of the effects of agricultural pesticide uses on peripheral nerve conduction in China. Sci. Rep. 8, 9621. https://doi.org/10.1038/s41598-018-27713-6

Reviewer 2 Report

The manuscript titled: “The toxic effects of glyphosate on the nervous system: a systematic review” is very interesting for readers. The subjects taken up by Authors is very important, because glyphosate is a toxic compound, which effect do not appear immediately after exposure to pesticide, as the authors noted (page 5, paragraph 2, line 2).

The review is well prepared, in it the influence of glyphosate on nervous system in rodents, humans and zebrafish (or other fish species) was described. The Authors included the figure (Figure 3), on which the mechanism of action of glyphosate in nervous system is presented, what significantly increases the quality of this paper. The Authors made a detailed analysis of the literature in described topic.

Despite the many advantages of reviewed manuscript, due to the duty of the reviewer, I present below comments to the work:

- Abstract – the Authors included in this section the same information in duplicate:

“…exposure to doses found in some environments has been shown to produce significant alterations in the function of the nervous system. (…) exposure to glyphosate produces important alterations in the structure and function of the nervous system of humans, rodents, fish, and invertebrates”. Please short these sentences.

- it should be better to present the ability of glyphosate to block shikimic acid pathway on figure (paragraph 1, page 2). It will help to understand this mechanism and make the readers easily to follow for them.

- Results section – please combine the subsection “Effect of glyphosate in zebrafish model” with the section “ Effect of glyphosate in other fish species”.

Author Response

  1. Abstract – the Authors included in this section the same information in duplicate: “…exposure to doses found in some environments has been shown to produce significant alterations in the function of the nervous system. (…) exposure to glyphosate produces important alterations in the structure and function of the nervous system of humans, rodents, fish, and invertebrates”. Please short these sentences.

Based on the reviewer's comment, we have removed the following sentence from the abstract so that it is not redundant: “in some cases, exposure to doses found in some environments has been shown to produce significant alterations in the function of the nervous system”.

  1. It should be better to present the ability of glyphosate to block shikimic acid pathway on figure (paragraph 1, page 2). It will help to understand this mechanism and make the readers easily to follow for them.

In response to the reviewer's recommendation, we have included a scheme showing the ability of glyphosate to block the shikimic acid pathway. Figure 1 shows the different steps that make up the shikimic acid pathway and indicates the point in the pathway where glyphosate acts.

  1. Results section – please combine the subsection “Effect of glyphosate in zebrafish model” with the section “Effect of glyphosate in other fish species”.

According to the reviewer's recommendation, we have combined the subsections “Effects of glyphosate in zebrafish model” and “Effects of glyphosate in other fish species”.
